# Computational enzyme design by catalytic motif scaffolding

Markus Braun[1,6], Adrian Tripp[1,6], Morakot Chakatok[1], Sigrid Kaltenbrunner[1], Celina Fischer[1], David Stoll[1], Aleksandar Bijelic[1], Wael Elaily[1], Massimo G. Totaro[1], Melanie Moser[1], Shlomo Y. Hoch[1,2], Horst Lechner[1], Federico Rossi[3], Matteo Aleotti[3], Mélanie Hall[3,4,5] & Gustav Oberdorfer[1,5 ✉]

Enzymes find broad use as biocatalysts in industry and medicine owing to their exquisite selectivity, efficiency and mild reaction conditions. Custom-designed enzymes can produce tailor-made biocatalysts with potential applications that extend beyond natural reactions. However, current design methods require testing a large number of designs and mostly produce de novo enzymes with low catalytic activities[1–3]. As a result, they require costly experimental optimization and high-throughput screening to be industrially viable[4,5]. Here we present rotamer inverted fragment finder–diffusion (Riff-Diff), a hybrid machine learning and atomistic modelling strategy for scaffolding catalytic arrays in de novo proteins. We highlight the general applicability of Riff-Diff by designing enzymes for two mechanistically distinct chemical transformations, the retro-aldol reaction and the Morita–Baylis–Hillman reaction. We show that in both cases, it is possible to generate catalysts that exhibit activities rivalling those optimized by in vitro evolution, along with exquisite stereoselectivity. High-resolution structures of six of the designs revealed near-atomic active site design precision. The design strategy can, in principle, be applied to any catalytically competent amino acid array. These findings lay the basis for practical applicability of de novo protein catalysts in synthesis and describe fundamental principles of protein design and enzyme catalysis.

Natural enzymes have emerged as an indispensable tool for chemists, offering unparalleled precision and selectivity in chemical transformations[6]. However, identifying natural enzymes with the desired activity can require extensive amounts of resources and screening capabilities. Recently, computational protein design has enabled the creation of biocatalysts designed for a variety of reactions[7–9]. Yet, an often under-emphasized limitation of current approaches to enzyme design is the low initial catalytic rates of the designed biocatalysts. Even with recent advances[10], the current paradigm for generating efficient biocatalysts is to compensate for low initial efficiencies with high-throughput screening and directed evolution[11]. Although this approach is valuable and robust, it is not suited to quickly create novel, proficient catalysts for chemical reactions that are not found in nature or are exceedingly difficult to access through high-throughput screening. Thus, it is necessary to advance our ability to design efficient enzymes using a one-shot approach.

De novo enzyme design strategies are commonly based on the transition-state model of enzyme catalysis[12]. The model proposes that functional groups in the active site accelerate chemical reactions by stabilizing the transition state of the reaction over the ground state. Following this model, minimal active sites, called theozymes, can be constructed by placing amino acid functional groups in a stabilizing geometry around a model of the transition state[13,14]. In the first

computationally designed enzymes, these theozymes were introduced into cavities of natural proteins, which could support the desired geometry[1,3,15,16]. Enzymes designed using this method successfully catalysed chemical transformations, but with low catalytic rates, and often required extensive screening[1–3]. Among the main limiting factors of the approach was the accuracy with which the catalytic sidechains could be placed within the active site. Successful designs exhibited sidechain root mean square deviation (RMSD) values of less than 1 Å between the geometry in the theozyme and a high-resolution experimental structure. Inactive designs, however, showed unproductive conformations[17–19]. The recent transformative successes in accurate structure prediction methods and protein design tools warrant another attempt to solve this problem[20,21].

Since the first successful attempts, critical analysis of designed enzymes and their evolved variants has revealed additional aspects of the enzyme design problem. One such challenge was the difficulty in predicting optimal theozymes for the reaction of interest[18,22,23]. This was reflected by early mutational studies of designed retro-aldolases, for which multiple catalytic interactions were programmed to stabilize the transition states of the reaction[3]. Of the programmed interactions, only the catalytic lysine buried in the hydrophobic binding pocket significantly contributed to catalysis. Directed evolution of one design replaced the initial catalytic machinery with a completely

[1]Institute of Biochemistry, Graz University of Technology, Graz, Austria. [2]Weizmann Institute of Science, Rehovot, Israel. [3]Institute of Chemistry, University of Graz, Graz, Austria. [4]BioHealth, Field of Excellence, University of Graz, Graz, Austria. [5]BioTechMed Graz, Graz, Austria. [6]These authors contributed equally: Markus Braun, Adrian Tripp. ✉e-mail: gustav.oberdorfer@tugraz.at

new catalytic tetrad[5]. Furthermore, molecular dynamics simulations of retro-aldolases at intermediate steps during directed evolution demonstrated that conformational populations of catalytic amino acids were increasingly rigidified towards catalytically productive conformations[24]. Similarly, conformational distributions of active site residues in designed Kemp eliminases were shown to discriminate between active and inactive designs[25–28]. In addition, backbone-to-sidechain compatibility and atomic density inside an active site pocket were found to be predictors of activity for a class of recombined de novo-designed xylanases[29].

In this study, we investigated whether efficient biocatalysts can be created in 'one shot' by incorporating three distinct catalytic centres in de novo protein backbones. We developed Riff-Diff, a method that focuses on the precise placement of backbone–sidechain fragments, the design precision of the given catalytic arrays, and the design of custom binding pockets. To test the effectiveness of our approach, we designed retro-aldol enzymes based on a previously reported, highly optimized catalytic tetrad[5] and Morita–Baylis–Hillmanases (MBHases) starting from two distinct evolved active site arrays[4,30]. Detailed biochemical, biophysical, and structural characterization, including molecular dynamics simulations, allowed us to test current hypotheses about the precision of the placement of active site atoms, and conformational dynamics in the designed enzymes. The extraordinary wealth of these data comes from the same catalytic environment placed into different backbones, allowing us to draw conclusions for de novo enzyme design that were not obvious before.

## Artificial motif libraries

We set out to create Riff-Diff as a computational pipeline that designs enzymes from scratch when given an array of amino acids as input. Previous attempts at grafting catalytic arrays into proteins have shown that the achievable catalytic effect of an array depends on how precisely it can be reproduced in the designed active site[7,18,26,31]. We utilized RFdiffusion[20] to scaffold the initial backbone around the input array and added modifications to improve design precision. First, each amino acid of the input array is embedded into a helical fragment. The rotamers of each amino acid are selected for compatibility with the fragment backbone to ensure a minimal energetic penalty is associated with the preferential catalytic rotamer[32]. We call the combination of the individual fragments an artificial motif.

The arrangement of helical fragments in an artificial motif depends on the positions of its functional group atoms and the chosen rotamers. This arbitrary arrangement of fragments can result in physically implausible motifs and low scaffolding success rates. To increase scaffolding success, we developed a script that generates libraries of artificial motifs from catalytic arrays from which high-quality motifs can be sampled. The script starts by inverting rotamers of the catalytic array, which fixes the functional groups in space and varies the position of their backbone atoms. Rotamers are selected to be compatible with the phi-psi angle combination of the fragment backbones, which are subsequently placed onto the rotamer backbone atoms. Next, a search for all possible combinations of fragments that are not sterically clashing with each other is performed. Non-clashing assemblies are ranked according to the rotamer probabilities of the individual active site residues and by their abundance in the Protein Data Bank (PDB). The identified assemblies are aggregated and stored in an artificial motif library (Fig. 1a). This library is then used as input to RFdiffusion to scaffold enzyme backbones.

## Ensuring formation of binding pockets

An essential feature of biocatalysts is their substrate specificity, which is dictated by well-defined interactions of the substrate with asymmetric binding pocket of the enzyme. To analyse how deep substrates are typically buried within binding pockets, we extracted an enzyme dataset from the PDBbind database[33]. In this dataset, an average of 7.1 α-carbons were within 8 Å of the substrate, normalized by the number of ligand-heavy atoms. We used this metric to evaluate ligand burial in the substrate pockets of backbones scaffolded by RFdiffusion. The RFdiffusion substrate auxiliary potential decreased the number of clashes between ligand and backbone but failed to promote the formation of well-defined substrate binding pockets. This resulted in a trade-off between clashes and substrate interactions in the scaffolded artificial motifs. To circumvent the trade-off, we implemented a novel approach to enforce pockets during diffusion by adding an α-helix as an entry-channel placeholder to each artificial motif at the position of the binding pocket. A custom auxiliary potential then places the centre of the denoising trajectory on the helix and enforces a distance constraint of all backbone atoms to this centre (Supplementary Methods). This ensures that the generated binding pockets exhibit characteristics that mimic those of natural enzymes (Fig. 1b). The 'placeholder' helix is removed after diffusion, leaving a vacant binding pocket.

## Riff-Diff scaffolds artificial motifs

An integral part of Riff-Diff is its backbone refinement protocol in which the diffused enzyme backbones are refined iteratively. The quality of de novo backbone–sidechain pairs improves when predicted structures are used for subsequent sequence design[34,35]. Therefore, we implemented iterative refinement cycles into Riff-Diff that bias the generated backbones towards the geometry of their artificial motif. Each cycle begins with LigandMPNN[36] threading sequences onto the backbones. FastRelax[37] then optimizes both backbone and active site orientations, guided by the coordinates of the artificial motif as constraints. Relaxed models are fed back into LigandMPNN for sequence redesign, and we use ESMFold[38] to predict the corresponding structures. The highest-scoring designs continue into the next refinement cycle.

Following backbone refinement, we utilize the Rosetta CoupledMoves protocol[39] to refine ligand interactions with the binding pocket. Alternatively, LigandMPNN can be utilized to generate the final enzyme sequences. In the final evaluation step, these sequences are predicted with AlphaFold2 and ranked with a combination of metrics for structure quality and active site positioning. We combined the scripts for generating artificial motif libraries, backbone diffusion and refinement, and CoupledMoves into a single method for the semi-automated design of enzymes, called Riff-Diff (Fig. 1c). Riff-Diff requires only minimal user input to generate sequences that scaffold a given catalytic array. The code, alongside detailed instructions for use, can be found at https://github.com/mabr3112/riff_diff_protflow. Detailed information on the protocol is provided in the Supplementary Methods.

## Design of de novo retro-aldolases

With the ability to build de novo scaffolds with binding pockets around arbitrary catalytic arrays, we investigated whether Riff-Diff could produce efficient enzymes when starting from an established catalytic array. As a model system, we selected the catalytic tetrad of the artificial retro-aldolase RA95.5-8F (PDB: 5AN7)[5], which catalyses the retro-aldol cleavage of (R)-methodol (1) to 6-methoxy-2-naphthaldehyde (2) and acetone (Fig. 1d,e). The product 2 of the reaction can be screened fluorescently and the tetrad is well-studied, providing a robust framework to assess de novo design performance[5,40,41].

Sequences for this tetrad designed by the Riff-Diff pipeline were generally predicted to fold as designed. Metrics for AlphaFold2 pLDDT, active site RMSD and Rosetta energy for the predicted structures can be found in Supplementary Fig. 1. To select designs for experimental

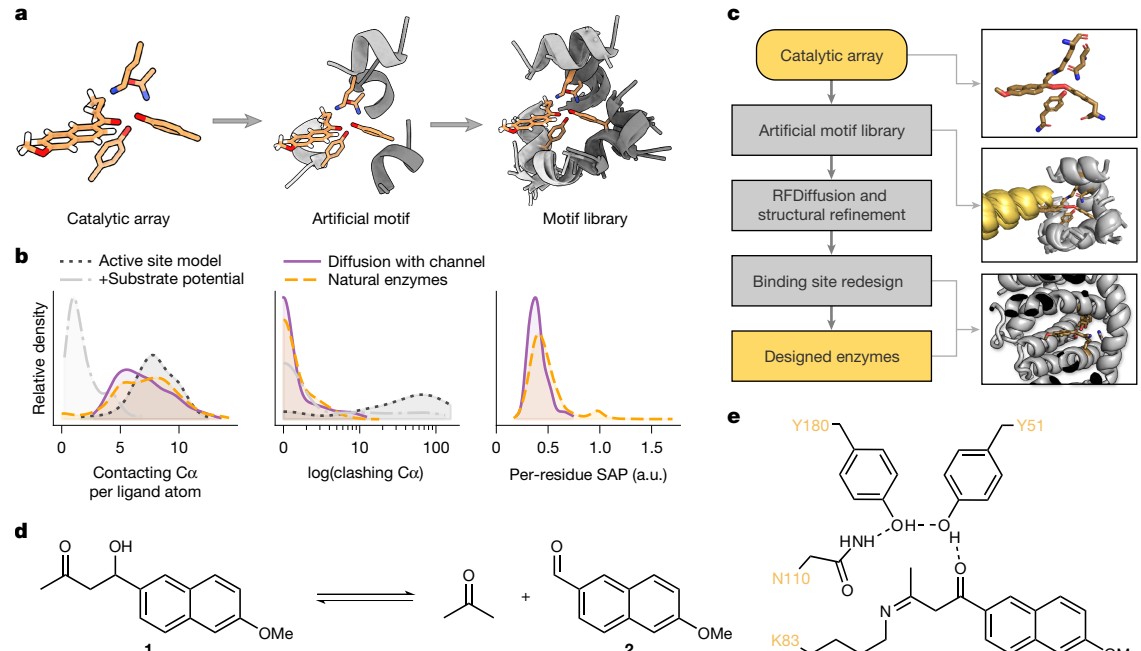

**Fig. 1 | Riff-Diff scaffolds de novo enzymes starting from catalytic arrays.** **a**, Artificial motif libraries are collections of artificial motifs that are constructed from arrays of sidechains. **b**, Binding pockets of natural enzymes (yellow) typically bury their substrates, measured by the number of α-carbons within 8 Å of the substrate. The RFdiffusion substrate potential offers only a trade-off between substrate burial and clashes (overlaps) with the pocket (light and dark grey). Riff-Diff (purple) scaffolds enzyme backbones that bury the substrate in binding pockets that are reminiscent of natural enzymes. Right, the SAP of the

designed enzyme is similar to that of natural enzymes. a.u., arbitrary units. **c**, Schematic overview of the semi-automated Riff-Diff pipeline. The channel placeholder helix is shown in yellow. **d**, A retro-aldol cleavage transforms methodol (**1**) to 6-methoxy-2-naphthaldehyde (**2**) and acetone. **e**, Four residues complete the catalytic array to catalyse the retro-aldol reaction (PDB: 5AN7). The lysine is covalently linked to the mechanistic inhibitor 1-(6-methoxynaphthalen-2-yl)butane-1,3-dione.

testing, we ranked the pool of designed enzymes predominantly by the sidechain RMSD of the catalytic tetrad. After additional manual inspection, we excluded designs for which the binding pocket appeared inaccessible to the substrate. The final selection consisted of 36 sequences derived from 12 unique backbones generated from 8 unique artificial motifs, named RAD1–36 (Supplementary Table 1). The backbones adopted novel folds, having an average maximum TM-score of 0.49 when searched with FoldSeek[42] against the PDB. In addition, no similar sequences could be found for any of the 36 sequences in a protein–protein BLAST search against the non-redundant protein sequence database[43]. The set of 36 designed retro-aldolases was ordered as synthetic genes, of which 35 could be cloned. An initial screen for expression and activity revealed that all 35 designs expressed soluble and 32 (91%) designs exhibited activity towards racemic methodol (*rac*-methodol) above the negative control. Notably, seven designs exhibited rates of product formation several orders of magnitude higher than the average conversion within the screen (Supplementary Fig. 2).

## Characterization of retro-aldolases

To investigate the origins of the divergent catalytic rates of the designs, we purified them from large-scale expressions, assessed foldedness and kinetic parameters, and investigated catalytic contributions of the tetrad amino acids. Gel filtration analysis showed monodisperse peaks corresponding to monomeric species (Fig. 2a) for all designs, and intact mass spectrometry validated the identity of the purified enzymes (Supplementary Table 2). Circular dichroism (CD) spectroscopy also confirmed their helical architecture (Supplementary Fig. 3). In addition, we collected small-angle X-ray scattering (SAXS) data to evaluate the overall foldedness of our enzymes in solution. To judge whether a protein roughly adopted the designed structure, we used dimensionless

Kratky plots (Supplementary Fig. 4), computed the expected radius of gyration, and calculated $\chi^2$ values for the fits between predicted and measured scattering data (Supplementary Table 3). Overall, SAXS measurements confirmed the expected fold for 29 of 35 designs and CD and gel filtration results confirmed all designs to express as soluble, α-helical monomers (Fig. 2b).

Next, we determined Michaelis–Menten parameters for 30 of the 35 produced retro-aldolases and compared their catalytic performance with previous designs and the highly evolved variant RA95.5-8F (Supplementary Fig. 5 and Supplementary Table 4). The catalytically most proficient designs were RAD35 and RAD29 (Fig. 2c). They catalysed the cleavage of *rac*-methodol with $k_{cat}$ (catalytic rate constant) values of $3.6 \times 10^{-2}\,\mathrm{s}^{-1}$ and $3.1 \times 10^{-2}\,\mathrm{s}^{-1}$, respectively, a roughly $5 \times 10^6$-fold rate acceleration over the uncatalysed reaction. Thus, our method produced enzymes orders of magnitude faster than previous computationally designed retro-aldolases, surpassing, for example, the activity of the evolved catalytic antibody 38C2 (refs. 44,45) ($k_{cat} = 1.1 \times 10^{-2}\,\mathrm{s}^{-1}$). A table of catalytic rates of past designed and evolved retro-aldolases was added to the supplementary materials for reference (Supplementary Table 5). Of particular note is the high affinity of RAD29 for *rac*-methodol. With a Michaelis constant ($K_m$) of around 100 μM, its catalytic efficiency of 290 $\mathrm{M}^{-1}\,\mathrm{s}^{-1}$ approaches that of the extensively evolved RA95.5-5 (ref. 44) (320 $\mathrm{M}^{-1}\,\mathrm{s}^{-1}$). We next sought to investigate the extent to which the tetrad residues participated in catalysis.

During the first steps of the reaction, the lysine of the tetrad attacks the carbonyl group of methodol to form a high-energy, hemiaminal intermediate (Supplementary Fig. 6). The central role of this lysine to catalysis in our enzymes was corroborated by the diminishing activities upon its substitution with alanine (Supplementary Fig. 2). For 22 designs, the catalytic rate ($<10^{-3}\,\mathrm{s}^{-1}$) falls within the range achievable by an isolated lysine in a hydrophobic pocket[5,24] (Fig. 2e). Seven designs

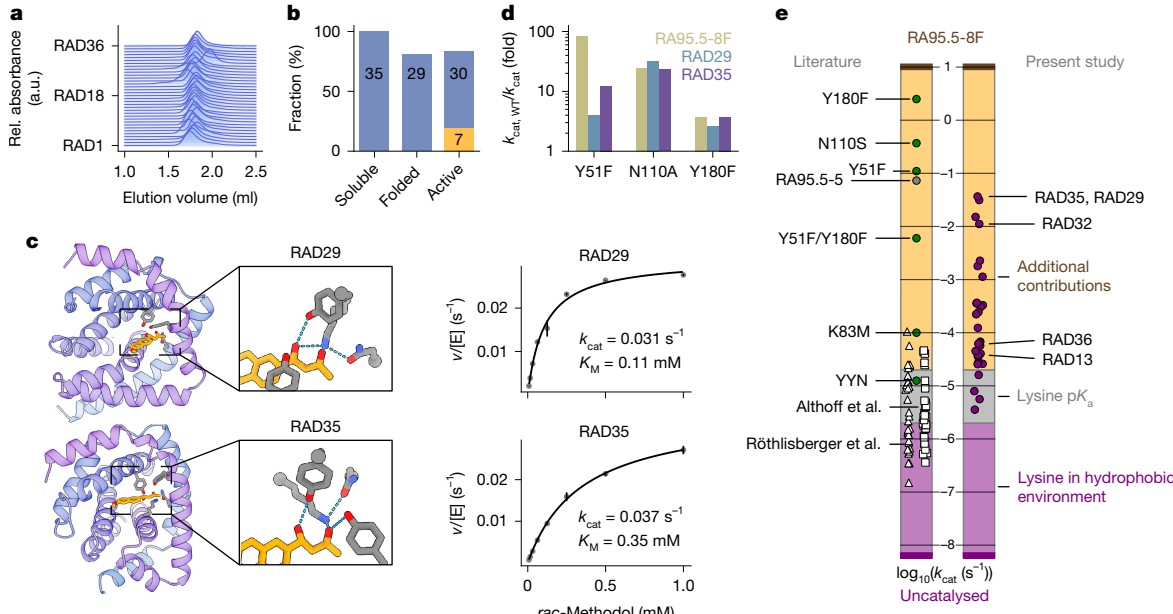

**Fig. 2 | Activities of designed retro-aldol enzymes exceeded those of previous one-shot designs. a**, All retro-aldolases eluted at the volume corresponding to the monomeric peak, as evidenced by size-exclusion chromatography. Size-exclusion chromatography traces were normalized and stacked. Rel., relative. **b**, Twenty-nine out of 35 designed retro-aldolases were correctly folded according to SAXS (FoXS $\chi^2$ <5). For 30 designs, product formation in the initial activity screen exceeded the background reaction. Seven designs exhibited $k_{cat} > 10^{-3} s^{-1}$ (yellow bar). **c**, Right, RAD29 and RAD35 exhibited the highest activity among the designed retro-aldolases. Error bars indicate standard deviations of triplicate measurements. Left, the designed structures in complex

with (R)-methodol, as predicted by AlphaFold3. **d**, Site-directed mutagenesis confirmed the participation of the tetrad residues in the catalytic mechanism. The residue numbering refers to the position of the catalytic residues in the model tetrad, not in the final designs. For asparagine-to-alanine variants, we compare the decrease in activity with RA95.5-8F(N110S). WT, wild type. **e**, For most enzymes, rate accelerations exceeded catalytic effects provided by the lysine in the hydrophobic pocket alone. Left bar, white triangles and squares represent computationally designed enzymes from previous design campaigns[15,22], and green dots represent variants that emerged after directed evolution. Right bar, computationally designed retro-aldolases in this study.

exceeded this range, suggesting that their tetrad residues might participate in the catalytic mechanism. To gain further insights, we determined Michaelis–Menten parameters for site-directed mutagenesis variants targeting the tetrad residues of our most active designs, RAD29 and RAD35 (Supplementary Fig. 7). The substitutions corresponding to N110A and Y180F in RA95.5-8F reduced $k_{cat}$ of both designs by approximately 2-fold and 20-fold, respectively, similar to the original tetrad. By contrast, the exchange Y51F only caused a moderate 3-fold (RAD29) and 10-fold (RAD35) decrease in $k_{cat}$ compared with the 100-fold decrease for the same exchange in RA95.5-8F (Fig. 2d).

The original tetrad requires its residues to occupy distinct protonation states throughout the reaction, making the reaction rate sensitive to the pH of its environment. We determined the pH rate dependency for several RAD designs and found a range of apparent p$K_{a1}$ values between 7.0 and 9.0 (Supplementary Fig. 8), higher than for the original tetrad (p$K_{a1}$ = 6.2). Notably, the pH rate profiles of RAD35 and RAD29 revealed that their kinetic parameters were determined below their pH optimum, leading to underestimated kinetic parameters. Together, the pH rate curves for RAD29 and RAD35 and the results from site-directed mutagenesis show the participation of the tetrad residues in catalysis.

## RADs are stereoselective biocatalysts

The catalytic rate of an enzyme is one among several metrics that define its utility as a biocatalyst. De novo-designed proteins are known for their high thermal stability, which offers an advantage over natural enzymes. CD melting curves confirmed that all but one of the designed retro-aldolases remained folded above 90 °C (Fig. 3a). To further investigate the thermodynamic stability of the designs, we measured chemical denaturation midpoints with guanidinium hydrochloride (GdnHCl). Twenty out of 33 enzymes displayed cooperative unfolding

with midpoints of denaturation ranging from 2.5 M to more than 6.5 M GdnHCl (Fig. 3b and Supplementary Fig. 9), with RAD29 and RAD35 being among the most stable. In addition, we found statistically significant ($P < 0.05$) correlations between observed denaturation midpoints and prediction confidence by AlphaFold2 (average pLDDT), the total score of the Rosetta energy function, core atomic density, and spatial aggregation propensity (SAP) score (Supplementary Fig. 10). Using these metrics, we built a simple linear regression model that predicted chemical denaturation midpoints with a Pearson's $R$ of 0.8 (Fig. 3c).

Determination of the turnover number of RAD29 and RAD35 resulted in 1,000 and 895 turnovers after reacting for 48 h in the retro-aldol direction (Fig. 3d). Stereoselectivity was assessed by-product formation in the aldol direction −the condensation of **2** with acetone to methodol. Whereas RAD29 was modestly stereoselective with an E-value of 4 towards the formation of (R)-methodol (60% enantiomeric excess), RAD35 displayed exquisite (R)-stereoselectivity and an E-value of over 200 towards (R)-methodol (99% enantiomeric excess, Fig. 3e). These values were corroborated by the analysis of the kinetic resolution of rac-methodol, with RAD29 and RAD35 displaying similar selectivity in the retro-aldol reaction. Under optimized reaction conditions, RAD29 and RAD35 converted 70% and 47.5% of the racemic substrate (Fig. 3f). The high stereoselectivity, turnover, and thermal stability make RAD35 a promising biocatalyst towards application under process-relevant conditions[6].

## Unexpected structure–function relation

We obtained crystal structures of the four designs: RAD13 ($k_{cat}$ = $3.8 \times 10^{-5} s^{-1}$), RAD17 ($k_{cat}$ not determined), RAD32 ($k_{cat}$ = $1.1 \times 10^{-2} s^{-1}$), and RAD36 ($k_{cat}$ = $6.2 \times 10^{-5} s^{-1}$). The divergent catalytic rates of the designs are ideal for investigating how much of the variation could be explained by sidechain packing in the active site. The crystal structures agree

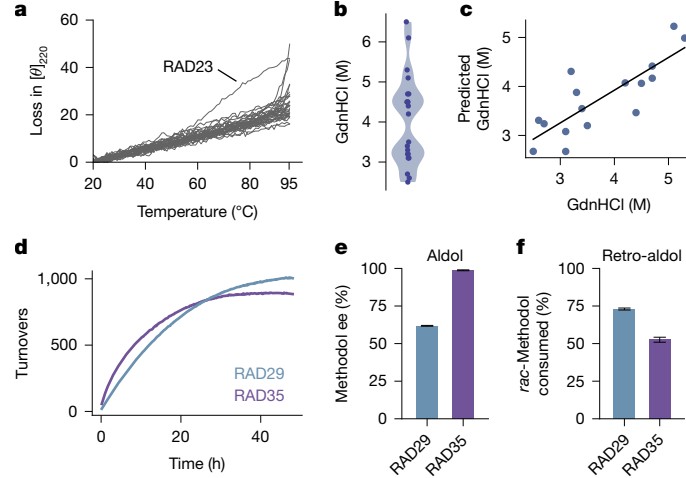

**Fig. 3 | Designed retro-aldolases are highly stable, enantioselective and can catalyse multiple turnovers. a**, CD melting curves confirmed high thermodynamic stability. Except for RAD23, only a negligible loss in signal intensity at 220 nm can be observed up to 95 °C. **b**, According to CD experiments, chemical denaturation midpoints for 20 of the 35 designs ranged from 2.5 M GdnHCl to more than 6 M. **c**, A linear regression model based on computational design metrics (Rosetta total score, average Alphafold2 pLDDT, spatial aggregation propensity and core contacts) can predict chemical denaturation midpoints with a Pearson's $R$ of 0.8. **d**, RAD29 and RAD35 can catalyse 1,000 and 895 turnovers, respectively. **e**, RAD29 and RAD35 exhibit stereoselectivity towards ($R$)-**1**, with enantiomeric excess (ee) of 60% and 99%, respectively. Bars correspond to the measured ee values of product ($R$)-**1** at 5% substrate conversion in the aldol direction. **f**, Consumption of *rac*-**1** after 24 h reaction time. RAD29 consumed 70% of *rac*-**1**, leaving 30% unreacted ($S$)-**1** with 77% ee. RAD35 consumed 47.5% of *rac*-**1**, leaving 52.5% unreacted ($S$)-**1** with 88% ee. Error bars in **e**,**f** indicate standard deviations of triplicate measurements.

with the design models, having backbone $C_\alpha$ RMSD values between 0.68 Å and 1.2 Å (Fig. 4a). The catalytic residues of RAD13 and RAD17 adopted the correct conformation with heavy-atom sidechain RMSD values of 0.42 Å and 1.09 Å to the design model and 0.89 Å and 1.2 Å to the catalytic geometry, respectively (Fig. 4b). Despite the near-atomic precision with which the tetrad was reproduced, RAD13 and RAD17 were more than $10^5$-fold slower than RA95.5-8F. This indicates that precise positioning of catalytic sidechains is by itself not sufficient to design efficient biocatalysts, even when scaffolding catalytic arrays that are derived from highly active enzymes.

In the active site of RAD36, we observed density for three distinct conformations of the catalytic lysine. One of the three lysine conformations points away from the active site, partially explaining its low catalytic rate. By contrast, in the active site of RAD32, the most active design for which an experimental structure could be obtained, a tyrosine of the catalytic tetrad (Tyr120) adopts a catalytically unproductive conformation. Another tyrosine, Tyr99, is positioned to compensate for this interaction, and a tenfold lower $k_{cat}$ of a Y99F variant confirmed its participation in the catalytic machinery. However, the exchange Y120F also led to a tenfold reduction in activity. The observed alternative conformation of Y120 concomitant with the reduced activity of Y120F hints at a potential conformational change induced by substrate binding. Both results highlight how the prediction of activity needs to account for catalytic contributions beyond the interactions specified in the catalytic array and their conformational dynamics.

## Flexibility as a barrier to activity

Flexibility in catalytic sidechains can reduce their likelihood of adopting active conformations, thereby slowing catalysis[24,28,46].

To evaluate the active site dynamics of our designs relative to RA95.5-8F, we computed 20 replicates of 20 ns apo-state molecular dynamics trajectories and used root mean square fluctuation as a metric to estimate sidechain flexibility. To evaluate the agreement between the simulated conformations and the catalytic geometry, we calculated two additional metrics: (1) the distance deviation of the hydrogen-bonding functional groups; and (2) the distance deviation of the active site functional groups after superimposition on the tetrad backbone atoms. Consistent with the literature, the active site of RA95.5-8F was rigid and remained positioned for catalysis throughout the trajectories. By contrast, all designed active sites were less consistently positioned than the original tetrad, except for isolated residues (Fig. 4c). In five designs, the catalytic lysine even drifted from the active site into a non-catalytic state. The generally more pronounced flexibility and alternative conformations of the active sites in the designed enzymes offer a further explanation for their lower activity compared with RA95.5-8F. However, neither of the calculated metrics from the molecular dynamics simulations correlated with activity, suggesting that the difference between designs might be linked to other factors.

To address the interactions of the active site with the bound substrate during the reaction, we used AlphaFold3 to model enzyme–substrate complexes directly. We predicted the designs complexed with ($R$)-methodol to assess the accessibility of the carbonyl group to the catalytic lysine and predicted the hemiaminal intermediate with a covalent link to the catalytic lysine. The predictions revealed the geometry of several catalytically relevant interactions: (1) the near-attack conformation of the lysine to the carbonyl group; (2) the hydrogen bonds to the alpha hydroxyl group; and (3) Tyr51 positioned between the two hydroxyl groups of the substrate to shuttle protons. Figure 4d depicts the predicted interactions for RA95.5-8F. Across all designs, we observed that substrate positioning and hydrogen bond distances of catalytic residues around the hemiaminal intermediate correlated with experimental activity. Notably, an active site metric derived from these features exhibited a Spearman $\rho$ of −0.64 with measured catalytic rates (Fig. 4d), outperforming apo-state metrics such as sidechain RMSD.

## Designs for a non-biological reaction

To demonstrate the general applicability of Riff-Diff to design enzymes, we selected the Morita–Baylis–Hillman (MBH) reaction as a second model system. In organic synthesis, this reaction—the carbon–carbon coupling of activated alkenes with aldehydes to yield functionalized allylic alcohols—is typically resource-intensive, requires high loading of nucleophilic catalysts and usually needs prolonged reaction times[47–49]. Although low levels of promiscuous MBH activity have been reported for several proteins[50–52], there are no known natural enzymes that catalyse this transformation efficiently. Computational design of MBHases based on an artificial cofactor embedded in streptavidin and cysteine or histidine residues as nucleophiles has resulted in low-activity designs[2,53]. However, directed evolution campaigns of one of these designs have revealed several highly active variants[4,30].

We used two catalytic arrays from evolved MBHase variants as examples, both of which catalyse the MBH reaction between 2-cyclohexen-1-one (**3**) and 4-nitrobenzaldehyde (**4**) (Fig. 5a). The first is the active site of BH32.14, an enzyme that emerged after 14 rounds of directed evolution[4]. It features a histidine (His23) as a nucleophile, a glutamic acid (Glu46) hydrogen bonding to the histidine Nε, and an arginine residue (Arg124). The arginine is central to catalysis in this catalytic array because it stabilizes negative charges on three oxyanion intermediates at the C1 and C3 positions that occur throughout the reaction trajectory and thus needs to adopt different rotameric states (Supplementary Fig. 11a). The second active site was taken from BH1.8, the

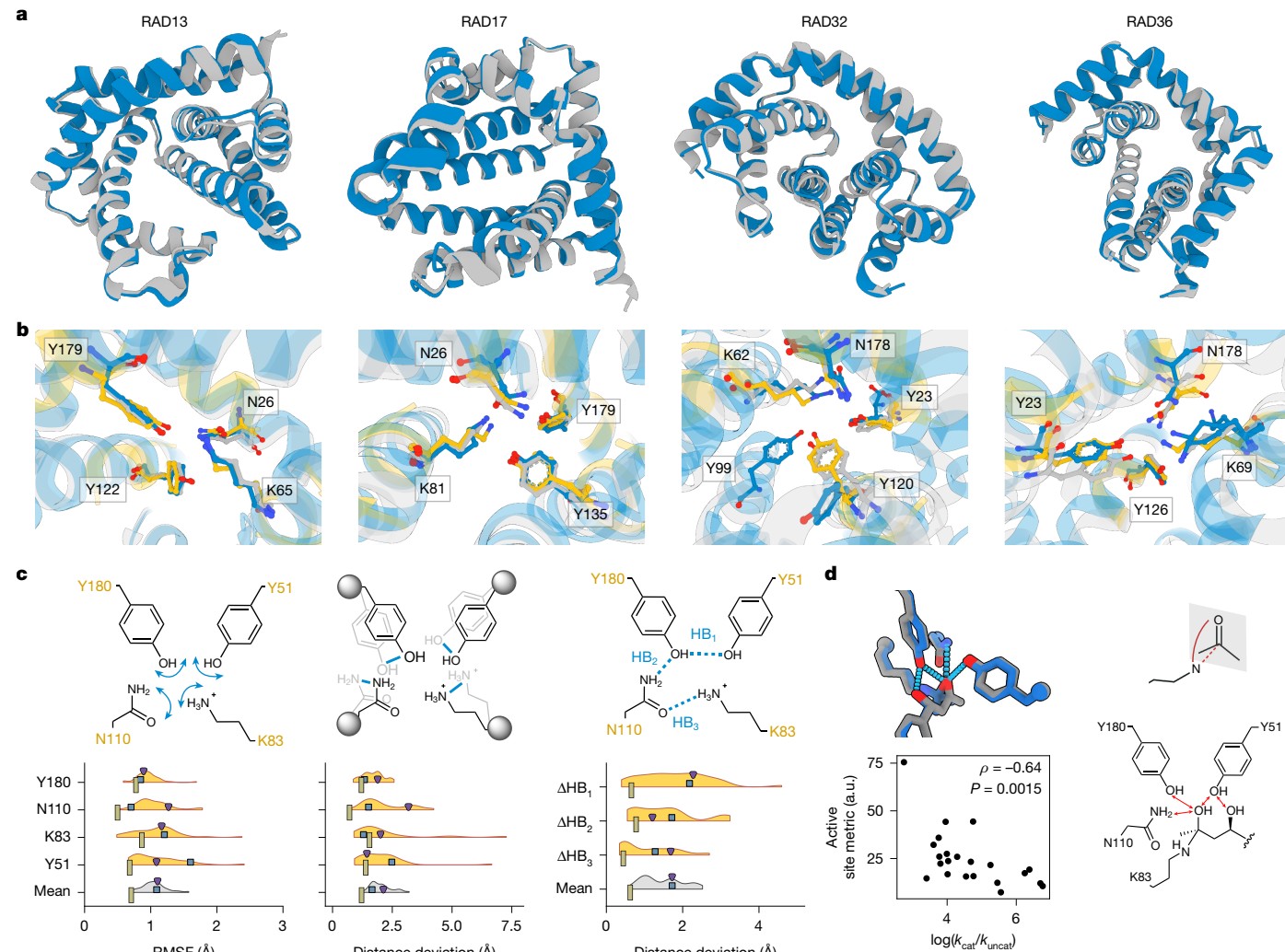

**Fig. 4 | Crystal structures of RAD designs reveal high accuracy of the scaffolded catalytic tetrad. a**, The backbones of the design models (grey) closely resemble the experimentally obtained crystal structures (blue), displaying overall $C_\alpha$ RMSD values below 1.2 Å. PDB IDs: 9GBT, 9FW5, 9FW7 and 9FWA. **b**, Active site residues in the crystal structures (blue) agree well with those in the design models (grey) and the tetrad (yellow). In the crystal structure of RAD32, the intended position of the tyrosine hydroxyl group is assumed by another tyrosine residue that was not part of the design model. Multiple conformations of the catalytic lysine residue are visible in the crystal structure of RAD36. The conformer with the highest occupancy is adopting a catalytically incompetent orientation. **c**, Individual metrics of active site rigidity: the root mean square fluctuation (RMSF) of all catalytic residues during molecular dynamics simulations (left), measured heavy-atom distances of the model to the catalytic array throughout the molecular dynamics trajectories (middle), and the resulting difference in H-bonding distances between the catalytic residues (right; $HB_1$–$HB_3$ are hydrogen bonds). Although the measured values are small, the designed active sites were still more flexible and less positioned than the original tetrad on average. Vertical olive rectangles correspond to RA95.5-8F, purple triangles correspond to RAD29, and blue squares correspond to RAD35. Amino acid indices indicate positions in RA95.5-8F. **d**, Bottom left, a composite metric calculated from complex predictions of AlphaFold3 correlated with activity. The composite metric combines interaction distances of the tetrad functional groups to the hemiaminal intermediate hydroxyl groups (top left) and the positioning of the substrate carbonyl group in the enzyme–substrate complex (top right). Bottom right, the catalytic array, depicting the measured distances. A value of $3.9 \times 10^{-7}$ $min^{-1}$ was used for $k_{uncat}$ (uncatalysed rate constant)[22].

MBHase with the highest activity to date[30]. This variant has emerged after eight additional rounds of directed evolution of BH32.8, following the substitution of the nucleophilic histidine with the noncanonical amino acid Nδ-methylhistidine. Its active site features a glutamic acid (Glu26) with a predicted $pK_a$ of 8.1, which was proposed to stabilize the negative charge on the C3 alkoxide in intermediate 2 and to mediate the rate-limiting proton transfer step from the C2 proton to the C3 alkoxide (transition state 4; Supplementary Fig. 11b).

We used Riff-Diff to create designs based on both active sites (Supplementary Methods) and selected 18 sequences (MBH1-18) derived from 14 unique backbones based on the active site array of BH32.14 (Fig. 5b) and 45 sequences (MBH19-63, 23 unique backbones) based on the active site of BH1.8 (Fig. 5c) for experimental characterization.

Sequences and computational metrics of ordered designs can be found in Supplementary Table 6 and Supplementary Fig. 12. Endpoint assays showed that 17 of the designs based on BH32.14 (94%) and 42 of the designs based on BH1.8 (93.3%) displayed product formation above background level and outperformed the small-molecule nucleophile catalysts imidazole and 4-dimethylaminopyridine (DMAP), emphasizing the robustness of the Riff-Diff design approach (Supplementary Table 7). Although median conversions in both design sets are similar, designs with the highest conversion are based on BH1.8 (Fig. 5d).

We determined Michaelis–Menten parameters for MBH18 and MBH48, the designs that showed the highest conversion after 8 h for their respective catalytic array (Supplementary Fig. 13). MBH18

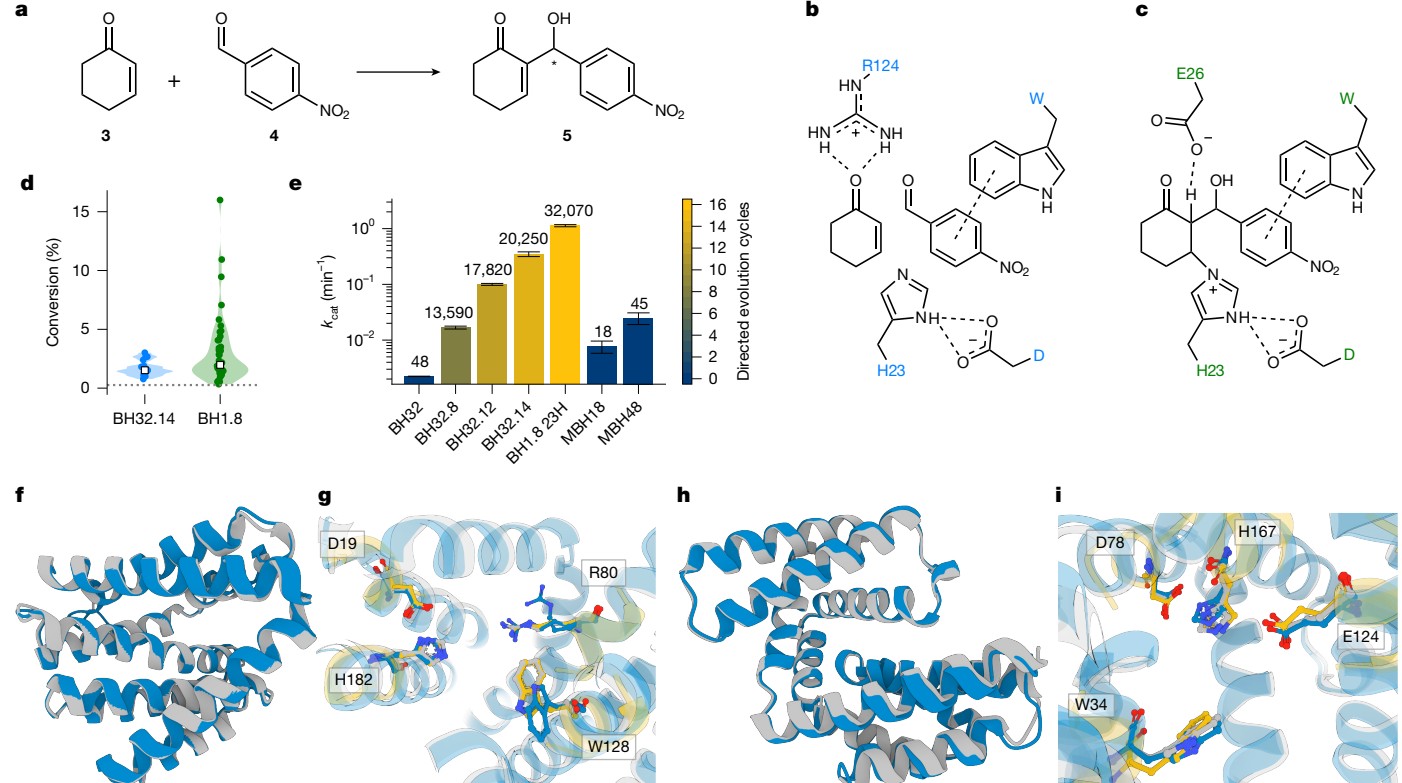

**Fig. 5 | De novo enzymes for the MBH reaction are active and agree with the design models. a**, Reaction scheme for the MBH reaction of 2-cyclohexenone (**3**) with 4-nitrobenzaldehyde (**4**) to yield 2-(hydroxy(4-nitrophenyl)methyl) cyclohex-2-en-1-one (**5**). **b,c**, Catalytic arrays based on transition state 1 from BH32.14 (**b**) and transition state 3 from BH1.8 (**c**). **d**, Conversion of substrates **3** and **4** after 8 h at 2 mol% catalyst loading for designs based on the active sites of BH32.14 and BH1.8. The dotted line marks the background reaction with lysozyme. **e**, The catalytic constant of MBH48 outperforms that of BH32.8, a variant that emerged after eight rounds of directed evolution. In BH1.8 23H, the noncanonical amino acid Nδ-methylhistidine was substituted with a regular histidine residue. Numbers above bars indicate the total number of screened clones; for evolved variants, this does not necessarily correspond to the number of unique sequences that were screened. Error bars indicate the 95% confidence interval. **f**, The crystal structure (blue) of MBH2 (PDB: 9QDP) agrees with the design model (grey), showing a Cα RMSD of 0.53 Å. **g**, In the active site of MBH2, the tryptophan sidechain in the crystal structure (blue) adopts a different conformation compared to the design model (grey) and the input geometry (yellow). Additionally, the catalytic arginine residue is present in two different conformations, one pointing away from the active site. **h**, The crystal structure of MBH48 (blue) (PDB: 9R7F) is in high agreement with the design model, exhibiting a Cα RMSD of 0.78 Å. **i**, Active site residues in the crystal structures of MBH48 (blue) agree well with those in the design model (grey) and the input geometry (yellow).

displays a $k_{cat}$ of $7.7 \times 10^{-3}$ min⁻¹, making it around 3.5-fold more active than BH32 ($k_{cat}$ of $2.2 \times 10^{-3}$ min⁻¹), the computationally designed starting point for directed evolution. A considerable amount of side product formation—a competing aldol reaction product according to similar previously reported high-performance liquid chromatography retention times[4,30]—can be observed for this design (Supplementary Fig. 14). With a $k_{cat}$ of 0.025 min⁻¹, MBH48 activity is 1.5 times higher than in BH32.8 ($k_{cat}$ of 0.0168 min⁻¹), a variant that emerged after screening 13,590 clones in 8 rounds of directed evolution (Fig. 5e). However, it remains 45-fold less active than BH1.8 23H ($k_{cat}$ of 1.13 min⁻¹), a BH1.8 variant with a regular histidine residue instead of Nδ-methylhistidine. Additionally, by-product formation in MBH48 is minimal (Supplementary Fig. 14). A comparison with Michaelis–Menten parameters for previously reported designed and evolved enzymes for the MBH reaction is presented in Supplementary Table 8.

## Structural characterization of MBHases

Experimental structure determination resulted in a 1.13 Å resolution crystal structure for MBH2, one of the designs displaying the lowest conversion. The structure agrees with the design model, showing an overall backbone Cα RMSD of 0.53 Å. The catalytic sidechains superimpose well with the catalytic geometry except for Trp140, which adopts a flipped conformation (sidechain RMSD 0.54 Å excluding Trp140, Fig. 5f,g). This flip widens the entrance channel and abolishes the designed π–π-stacking interactions with the substrate **4**. In addition, Arg92 adopted two conformations: one resembling the design model, the other pointing away from the active site. Together, these observations provide a possible reason for the low conversion rates of MBH2.

CD experiments confirmed the expected helical fold of MBH48, the most active MBHase design, with an unfolding onset at around 85 °C (Supplementary Fig. 15). The crystal structure of MBH48 agrees with its design model (Cα RMSD 0.78 Å; Fig. 5h), and the desired catalytic geometry was reproduced with a sidechain RMSD of 1.07 Å (Fig. 5i). Such precision might explain the high catalytic activity, however, our results show that high activity cannot be inferred from precise reproduction of the apo-state alone. Gaining a deeper understanding of the rate enhancement of the MBH reaction will require more than the two crystal structures presented here. To this end, we see the ability of Riff-Diff to scaffold catalytic arrays as a promising approach to study catalytic interactions in isolation from the complex environment of their natural enzymes.

## Conclusions and outlook

In one shot, Riff-Diff designed enzymes to catalyse (retro-)aldol and MBH reactions starting from catalytic arrays as inputs. The designed

enzymes exceed the activities achieved by previous design campaigns and rival those subjected to several rounds of directed evolution[4,22,31], while screening as few as 35 sequences in one instance and fewer than 100 overall. Our study demonstrated that Riff-Diff can design enantioselective enzymes with thermal stabilities of typical de novo scaffolds. The ability of Riff-Diff to design enzymes without experiment–design cycles constitutes a significant advance towards minimizing laborious high-throughput screens. The datasets generated in this study enabled us to detail reasons for higher and lower catalytic proficiencies and are an ideal test set to benchmark models of catalytic activity. We emphasize that de novo design campaigns like these for a diverse set of enzyme-catalysed reactions will complement the analysis of natural enzymes towards building general models of enzyme catalysis.

A critical challenge that remains in one-shot enzyme design is the accurate prediction of enzymatic activity before experimentation. This was highlighted by the negligible correlation between predicted enzyme rankings and actual activity in this and previous studies[8,10]. Yet, biochemical and computational analyses of our designed enzymes exposed limitations of established enzyme design strategies. One is the use of metrics for structural similarity, such as RMSD, to rank and optimize designed enzymes. RMSD is a popular metric for measuring how closely a designed active site aligns with the intended catalytic array, but it is agnostic to the geometric constraints of catalytic interactions. This could explain the discrepancy between the low RMSD in the active site of RAD13 and low catalytic activity and general poor predictiveness of sidechain RMSD for activity. Another limitation is the lack of control over conformational dynamics of established sequence design methods. Increased flexibility and mispositioning in the active sites of the retro-aldolases might have reduced their overall activity. Conversely, for the catalytic array of BH32.14, we could not specify the required flexibility of the catalytic arginine relative to the remaining catalytic residues. Replicating the precise control of natural enzymes over the dynamic behaviour of their active sites may hinge on the integration of computationally efficient tools that model conformational dynamics explicitly.

Finally, an essential factor that contributed to the high activity of our enzymes was the use of catalytic arrays extracted from efficient enzymes. For novel reactions, these catalytic arrays will need to be constructed from scratch—a challenge that requires a robust mechanistic understanding of how functional groups confer catalytic effects. We envision Riff-Diff as a tool that enables building and testing such knowledge. With the ability to robustly scaffold catalytic arrays, Riff-Diff offers a way to go beyond screening campaigns and help transform enzyme design from an orphan field into an approach that the broader biotechnological community can apply.

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

## Methods

### Cloning, protein production and purification

RAD and MBH genes were obtained from IDT and GenScript, respectively (sequences can be found in Supplementary Table 1). All genes were cloned into a vector containing a hexa-histidine-tag and TEV-cleavage site using Golden Gate assembly with a BsaI restriction enzyme[54]. The plasmid sequence is available at https://doi.org/10.5281/zenodo.15494922. Enzymes for amplification, mutagenesis and cloning were sourced from New England Biolabs.

For initial screening, the plasmids were transformed into *Escherichia coli* BL21 (DE3) STAR cells using standard procedure (30 min incubation on ice, 30 s heat shock at 42 °C, and regeneration in SOC medium at 37 °C, 1,000 rpm). A 96-well plate containing 500 µl of LB medium with 50 µg ml⁻¹ kanamycin in each well was inoculated with a single colony for each design, including a positive control, a negative control and a sterile control (sequences in Supplementary Table 1). Cultures were grown overnight at 37 °C, 250 rpm. One hundred microlitres of these cultures was used to inoculate 24-well plates containing 3 ml of ZY autoinduction medium. For each sample, two replicates were performed. Cultures were grown at 37 °C, 200 rpm for 6 h, then at 18 °C, 200 rpm for 18 h.

Samples were collected by centrifugation (20 min, 4,800$g$) and resuspended in lysis buffer (20 mM sodium phosphate (sodium phosphate), 500 mM NaCl, 1% $n$-octyl-β-$D$-glucopyranoside, 20 µg ml⁻¹ DNase I, 250 µg ml⁻¹ lysozyme, 1× cOmplete protease inhibitor pill per 100 ml, pH 7.4). Lysis was performed at room temperature for 1 h on a vibrating plate at 1,000 rpm. The lysate was clarified (20 min at 4,800$g$), and the supernatant was purified in a 96-well plate using magnetic Ni-NTA beads and an Opentrons OT-2 pipetting robot. The lysate was incubated with the magnetic beads for 20 min at room temperature, and the supernatant was discarded. Beads were washed with 200 µl wash buffer (20 mM sodium phosphate, 500 mM NaCl, 2 mM tris(2-carboxyethyl)phosphine (TCEP), pH 7.4) once. The supernatant was discarded, and bead-bound proteins were eluted by adding 100 µl elution buffer (20 mM sodium phosphate, 500 mM NaCl, 250 mM imidazole, 2 mM TCEP, pH 7.4) twice per well, yielding 200 µl eluate for each replicate. Protein concentration was determined via Bradford assay (bovine serum albumin standard curve). Protein purity was confirmed using SDS–PAGE.

Active site lysine-to-alanine variants were generated by site-directed mutagenesis using primers designed with the NEBaseChanger tool. PCR reactions contained 10 µl of 5× NEB Q5 buffer, 32.5 µl ddH₂O, 1 µl of dNTP mix (10 mM in ddH₂O), 2.5 µl of forward and reverse primers (10 µM in ddH₂O), 1 µl of template DNA (2 ng µl⁻¹) and 0.5 µl Q5 DNA polymerase. Reactions were cycled for 25 rounds (98 °C denaturation, primer-specific annealing temperature, 72 °C extension), with initial denaturation (30 s) and final extension (2 min) steps. A complete list of primers and annealing temperatures used can be found in Supplementary Table 9. To 1 µl of the PCR reaction, 1 µl each of T4 DNA ligase buffer, T4 DNA ligase, T4 polynucleotide kinase, and DpnI as well as 5 µl of ddH₂O were added and incubated at 22 °C for 1 h. Plasmids were transformed into *E. coli* TOP10 cells and plated on agar plates (50 µg kanamycin per ml agar). Plasmids from single colonies were isolated using a Monarch Spin Plasmid Miniprep kit and sequence-verified. Screening of lysine-to-alanine variants was performed in the same way as the original designs.

For batch production, 10 ml of TB medium containing 100 mg l⁻¹ kanamycin was inoculated with a single colony of BL21 (DE3) STAR cells containing the respective plasmid. After overnight growth at 37 °C, 140 rpm, 10 ml of the culture was used to inoculate 1 l TB medium (same antibiotic). Cultures were grown to an OD₆₀₀ of 0.6–0.8 at 37 °C and 140 rpm, and induction was initiated by adding isopropyl β-$D$-thiogalactopyranoside (IPTG) to a final concentration of 0.1 mM. Cells were collected 4–5 h after induction via centrifugation (20 min,

4,000$g$). For MBH19-63, cultures were incubated at 20 °C, 140 rpm overnight following induction and collection the next morning. Pellets were washed once with 30 ml of 0.9% NaCl solution at room temperature and stored at −20 °C.

Pellets were thawed and resuspended in lysis buffer (20 mM sodium phosphate, 500 mM NaCl, and a spatula of DNase I and lysozyme per 200 ml, pH 7.4). The suspensions were sonicated for 15 min on ice and the lysate was centrifuged at 43,000$g$ for 40 min. The supernatant was loaded onto gravity columns containing 1–2 ml nickel immobilized metal affinity chromatography (Ni-IMAC) resin equilibrated with lysis buffer and washed with wash buffer (see above). The purified proteins were eluted using an elution buffer (see above). Buffer was exchanged to storage buffer (20 mM sodium phosphate, 300 mM NaCl, 2 mM TCEP, pH 7.4 for RAD designs; 20 mM sodium phosphate, 150 mM NaCl pH 7.4 for MBH designs) using centrifugal filters. For RAD designs, His-tag cleavage was performed by adding 0.062 mg of TEV protease (produced in-house) per mg of protein and incubation at 4 °C overnight. The cleaved tag was removed using reverse Ni-IMAC. MBH His-tags were not removed. The final purification step consisted of gel filtration on a S75 Increase 10/300 GL or S75 10/300 column equilibrated with the respective storage buffer. Protein concentrations were determined by specific absorbance at 280 nm and Bio-Rad assay. Samples were flash-frozen in liquid nitrogen and stored at −80 °C.

### Intact mass spectrometry

Five microlitres of protein samples (10 µM) in RAD storage buffer were desalted on a Shim-pack Scepter C4-300 (G) column (3 µm) by washing with 1% methanol in the presence of 0.1% formic acid. Increasing concentrations of acetonitrile (MeCN, 1–95%) with 0.6% formic acid eluted the proteins into an Impact II ESI-Q-TOF (Bruker) mass spectrometer. Protein signatures were integrated and deconvoluted using the DataAnalysis maximum entropy function.

### Circular dichroism

CD and thermal denaturation experiments for retro-aldolase designs were performed on a JASCO-1500 CD-spectrophotometer in 10 mM sodium phosphate buffer, pH 7.4, containing 150 mM NaF and 0.5 mM TCEP at approximately 0.25 mg ml⁻¹ protein concentration. Spectra were recorded in 1 mm quartz cuvettes with a cap. Thermal denaturation was performed at 3 °C min⁻¹ while monitoring CD signal intensity at 220 nm. Spectra (190–260 nm) were recorded at 20 °C, 45 °C, 70 °C and 95 °C. Additional spectra were recorded after cooling down to 20 °C. Each spectrum consisted of three accumulations.

Chemical denaturation experiments were performed in 100 mM sodium phosphate buffer pH 7.4 containing 300 mM NaF and 1 mM TCEP at a final protein concentration of 0.6 mg ml⁻¹. GdnHCl from a 7.4 M stock solution (concentration confirmed via refractometry) was added to final concentrations between 0 M and 7.1 M. Protein samples were incubated at room temperature overnight, and the CD signal at 220 nm was recorded in quartz capillaries. Denaturation midpoints were calculated from a sigmoidal fit with the Python SciPy library.

MBH48 CD experiments were performed on an Applied Photophysics Chirascan V100 CD-spectrophotometer in 20 mM sodium phosphate buffer, pH 7.4, containing 150 mM NaF at a protein concentration of 0.1 mg ml⁻¹. Spectra were recorded in a 1 mm quartz cuvette with a cap. Thermal denaturation was performed at 1.5 °C min⁻¹, with full spectra recorded every 1 °C. An additional spectrum was recorded after returning to the starting temperature.

### SAXS

SAXS profiles were recorded at the ESRF BM29 BioSAXS beamline[55]. Samples were prepared in 20 mM sodium phosphate buffer, pH 7.4, containing 150 mM NaCl, 1 mM TCEP and 3% glycerol, at a protein concentration of 2–4 mg ml⁻¹. Blank measurements were performed using buffer from the flowthrough of centrifugal filters for each sample.

Raw data were processed using the ATSAS software package[56]. Frames were manually inspected, averaged and background subtracted. All parameters listed in Supplementary Table 2 were calculated using autorg/autognom functions in ATSAS. Low-$q$ trimming was guided by autorg recommendations. Scattering profiles were fit with design models or available crystal structures using FoXS[57], with offset correction and explicit hydrogens enabled, using $q$ values up to 0.5. C1 and C2 parameters were set to be flexible between 0.99 and 1.05 and −2 and 4, respectively.

## X-ray crystallography

Crystallization drops were set up with commercial crystallization screens using the vapour-diffusion method, using a mosquito Xtal3 crystallization robot (SPT Labtech) and incubated at 20 °C. The protein concentration varied between 10–30 mg ml$^{-1}$ in 20 mM sodium phosphate, 150 mM NaCl, 1 mM TCEP, pH 7.4 (RADs) or 20 mM HEPES, 150 mM NaCl, pH 7.4 (MBHase). The drop volume was 400 nl, with a 1:1 protein to precipitant solution ratio. Crystallization drops were equilibrated against 40 μl of precipitant solution. Crystals of RAD32 and MBH2 were obtained from manually set-up drops of 2 μl with 80 μl of precipitant solution in the reservoir. Depending on the design, crystals appeared after 1–14 days. Crystallization conditions and data collection and refinement statistics are provided in Extended Data Table 1. The obtained crystals were collected with CryoLoops (Hampton Research) and cryo-protected in mother liquor containing 25% glycerol or 25% PEG400, followed by flash freezing in liquid nitrogen. Diffraction data were collected at 100 K on ESRF beamlines (Grenoble, France). Complete datasets (360°) were collected to 2.43 Å (RAD13), 2.9 Å (RAD17), 2.0 Å (RAD32), 1.73 Å (RAD36), 1.13 Å (MBH2) and 1.93 Å (MBH48) resolution.

The collected data were processed using XDS[58] with the provided input file from the beamline. Data resolution cutoffs were determined by pairef[59]. Structure determination was performed by molecular replacement using PHASER[60] with the design models as search templates. The best solution was refined in reciprocal space with PHENIX[61] with 5% of the data used for $R_{free}$ and by real-space fitting steps against $\sigma A$-weighted $2F_o - F_c$ and $F_o - F_c$ electron density maps using COOT[62]. For RAD17, feature-enhanced maps[63] were generated to facilitate the modelling of sidechains of the active site residues (Supplementary Fig. 16).

## Activity measurements

Catalytic activity in the retro-aldol reaction was determined by following the formation of the aldehyde product **2** via measuring fluorescence emission at 452 nm at an excitation wavelength of 330 nm. Reactions were carried out in reaction buffer 1 containing 20 mM sodium phosphate, 300 mM NaCl, and 1 mM TCEP in 5 vol% dimethyl sulfoxide (DMSO), pH 7.4 at a 200 μl reaction volume in a 96-well-plate format at a temperature of 29 °C. Michaelis–Menten kinetics were measured with eight 1:1 serial dilutions starting at a concentration of 1 mM to a final concentration of 0.0078 mM *rac*-methodol. Parameters $k_{cat}$ and $K_M$ were determined by fitting reaction velocity and substrate concentration in a Michaelis–Menten model using the Python library SciPy, v.1.13.1.

pH profiles for RAD designs were determined at 50 μM *rac*-methodol concentration in 384-well plates at a total sample volume of 100 μl and a temperature of 25 °C. Final enzyme concentrations were set to 2 μM (RAD13), 12.6 μM (RAD17), 0.03 μM (RAD29), 0.06 μM (RAD32), 0.03 μM (RAD35) and 3.6 μM (RAD36). To 10 μl of protein samples in 10 mM sodium phosphate, 300 mM NaCl, pH 7.4 90 μl of reaction buffer 2 at various pH values (100 mM phosphate, 100 mM borate, 100 mM acetate buffer containing 55.56 μM *rac*-methodol and 5.56 vol% DMSO) were added. The pH of reaction buffer 2 was adjusted to values between 4.6 and 10.6 in 0.4 steps. $k_{cat}/K_M$ was determined via monitoring of fluorescence as described above. For each measurement, the background reaction at the respective pH was subtracted. p$K_a$ values were obtained according to a fit with a two-p$K_a$ model. Detailed information on software and equations can be found in the Supplementary Methods.

The aldol addition reaction was performed using a final enzyme concentration of 5 μM in reaction buffer 3 (20 mM sodium phosphate, 300 mM NaCl, pH 7.4). 5 mM of **2** was added from a stock solution in acetonitrile (MeCN, 15 vol% final concentration) followed by 5 vol% acetone. The reactions (final volume 500 μl) were incubated for 24 h at 30 °C and 120 rpm. Control reactions were run in the absence of enzymes. All reactions were performed in triplicate. The products were extracted using ethyl acetate (2× 250 μl) spiked with 0.5 vol% acetophenone as internal standard. The combined extracts were dried over anhydrous sodium sulfate and analysed by chiral-phase high-performance liquid chromatography (HPLC) as indicated below. The results were corrected for the background reaction under the same conditions. The identity of the compounds was further confirmed by gas chromatography–mass spectrometry and the use of authentic reference material. Substrate consumption was measured using a calibration curve.

For conversion and substrate enantiomeric excess determination in the retro-aldol reaction, a final enzyme concentration of 20 μM in reaction buffer 3 was used to catalyse the reaction with 2 mM *rac*-**1** as the substrate, added from a stock solution in MeCN (15 vol% final concentration). The samples were incubated for 24 h at 30 °C and 120 rpm. All experiments were performed as technical triplicates (for biological replicate measurements see Supplementary Table 4). The reaction was analysed using the same method as the forward aldol reaction, monitoring the consumption of both enantiomers and the formation of the aldehyde product **2**.

HPLC analyses were performed on a Shimadzu system (DGU-20A On-line Degasser, LC-20AD pump, SIL-20AC autosampler, CBM-20A system controller, SPD-M20A photodiode array detector, Shimadzu CTO-20AC column oven). The samples (5 μl) were analysed with an isocratic flow according to the following methods:

For elution order and absolute configuration: analysis on a Daicel Chiralpak IB (250 mm, ID 4.6 mm, particle size 5 μm) using *n*-heptane/2-propanol 90:10 (isocratic, flow rate of 1 ml min$^{-1}$, 30 °C, wavelength 254 nm); retention times: (*S*)-methodol 12.6 min, (*R*)-methodol 13.2 min (ref. 64) (Supplementary Fig. 17a,c).

For enantiomeric excess determination, analysis was performed on a Daicel Chiralcel OD-H (250 mm, ID 4.6 mm, particle size 5 μm) using *n*-heptane/2-propanol 92:8 (isocratic, flow rate of 1 ml min$^{-1}$, 30 °C, wavelength 254 nm), as complete baseline separation was achieved on an OD-H column. Retention times: (*S*)-methodol 18.5 min, (*R*)-methodol 20.0 min (see Supplementary Fig. 17b,d).

RAD turnover number experiments were performed at 0.1 μM final enzyme concentration and 2 mM final substrate concentration (*rac*-**1**) in reaction buffer 3 containing 15 vol% DMSO. Samples were incubated for 48 h at 29 °C. All experiments were performed in triplicate. Product formation was monitored using fluorescence and comparison to a calibration curve. The background reaction of the substrate without enzyme was subtracted.

Conversion screening of MBH designs was performed at an enzyme concentration of 100 μM. Reactions were performed in 96-microwell plates in 20 mM phosphate buffer pH 7.4 with 150 mM NaCl and 10 vol% DMSO at substrate concentrations of 25 mM **3** and 5 mM **4**. Samples were taken after 8 h of incubation at 40 °C and 800 rpm by quenching 10 μl of the reaction mixture with 10 μl of MeCN and subsequently used for HPLC analysis.

Michaelis–Menten kinetics for the reaction of **3** with **4** were recorded at 60 μM (MBH48) and 90 μM (MBH18) concentration in 20 mM sodium phosphate, pH 7.4, containing 150 mM NaCl and 10 vol% DMSO at 100 μl reaction volume in polypropylene 96-microwell plates. For reactions versus **3**, the concentration of **4** was fixed at 5 mM, whereas that of **3** ranged from 0.5 mM to 32 mM. For reactions versus **4**, the concentration of **3** was fixed at 25 mM, whereas that of **4** ranged from 0.1 mM to 6.4 mM. Reaction progress was sampled every 50 min (MBH48) or 2 h (MBH18) by quenching 10 μl of the reaction mixture with 10 μl of MeCN and subsequently analysed by HPLC. All reactions were performed in

triplicate. For MBH48, biological replicates were performed as well, with values for $K_M$ and $k_{cat}$ within the standard deviations of the first measurement. Initial reaction velocities ($V_0$) at each substrate concentration were determined from linear fits of conversion versus time. Information on software and equations is provided in the Supplementary Methods. Fits for individual measurements versus **3** and **4** are shown in Supplementary Fig. 13.

Conversion and kinetic measurements for MBH designs were analysed using a ThermoScientific UltiMate 3000 HPLC equipped with a Gemini SecurityGuard 4×2.0 cartridge and a Kinetex 5 µm XB-C18 100 Å column (50 × 2.1 mm, Phenomenex) at a flow rate of 1 ml min$^{-1}$. An isocratic method using 22 vol% MeCN in water at 20 °C was used for all measurements. The 2-(hydroxy(4-nitrophenyl)methyl)cyclohex-2-en-1-one (**5**) product retention time and concentration were determined via comparison to a calibration curve prepared from a chemically synthesized standard.

## Molecular dynamics simulations

Molecular dynamics simulations were performed using GROMACS[65] version 2023.4. The GRO coordinates of the design models and the topology files were generated using the pdb2gmx tool with the amber99sb-ildn force field. The molecular dynamics unit cell was defined as a dodecahedron with a minimum distance of 1.0 nm between the complex and the box edges using the gmx editconf tool. The system was solvated with the SPC/E water model using the gmx solvate tool and neutralized by adding Na$^+$ ions using the gmx grompp and gmx genion tools with a salt concentration of 150 mM. The system was then subjected to standard energy minimization and equilibration using the gmx grompp and gmx mdrun tools. Energy minimization was performed using the steepest descent algorithm with a maximum force of 1,000 kJ mol$^{-1}$ nm$^{-1}$ and a maximum of 5,000 steps. The equilibration consisted of two phases: NVT (constant number of particles, volume and temperature) and NPT (constant number of particles, pressure and temperature). The NVT phase was run for 100 ps with a temperature coupling of 298 K using the v-rescale thermostat. The NPT phase was run for 200 ps with a pressure coupling of 1 bar using the Parrinello–Rahman barostat. Backbone C$_\alpha$ RMSDs were followed throughout the trajectories to evaluate simulation stability (Supplementary Fig. 18).

Prior to the 20 ns replicate simulations, individual 50 ns simulations were run to confirm proper equilibration. The parameters for both runs were as follows: the temperature and pressure were set at 298 K and 1 bar by the v-rescale thermostat and Parrinello–Rahman barostat, respectively; hydrogen bonds were constrained using the LINCS algorithm; the Verlet cutoff scheme was used to process intra-atomic interactions; the PME method was implemented to account for Coulombic and Lennard–Jones interactions; and a van der Waals cut-off radius of 1.0 was applied. The simulations were analysed using the MDAnalysis Python package version 2.8.0 (ref. 66).

## Synthesis

All reagents used for synthesis were obtained from Sigma-Aldrich (except for pyridine, obtained from ThermoFisher Scientific) with min. 98% purity and used without further purification. Solvents were of HPLC grade. NMR spectra were measured on a Bruker Avance III 300 MHz NMR spectrometer. Chemical shifts are reported in ppm relative to TMS ($\delta$ = 0.00 ppm) and the coupling constants ($J$) in Hertz (Hz).

### *rac*-Methodol (*rac*-1) synthesis

6-Methoxy-2-naphthaldehyde **2** (500 mg, 2.69 mmol, 1 equivalent) was added to a 1:4 mixture of acetone (5.4 ml) and aqueous phosphate solution (22 ml, 10 mM NaH$_2$PO$_4$, 111 mM NaCl, 2.7 mM KCl in water, pH 7.4) (1:4). L-proline (62.2 mg, 0.2 equivalent) was added to the solution, and the reaction was stirred at room temperature for 48 h. Since the reaction had not sufficiently proceeded based on thin-layer chromatography monitoring, more acetone (5.4 ml) and L-proline (62.2 mg) were added

after 48 h, and the reaction was kept stirring for another 48 h, after which conversion appeared complete on thin-layer chromatography. Purification by flash chromatography (cyclohexane/ethyl acetate, 2:1) and evaporation of the solvent under reduced pressure yielded the final racemic product methodol (4-hydroxy-4-(6-methoxy-2-naphthalenyl)-2-butanone **1**) as a white solid (400 mg, 1.64 mmol) in 61% yield. The spectral data were in accordance with the literature[67] (Supplementary Figs. 19 and 20). $^1$H NMR (300 MHz, CDCl$_3$) $\delta$ 7.76–7.68 (m, 3H), 7.42 (dd, $J$ = 8.5, 1.9 Hz, 1H), 7.19–.08 (m, 2H), 5.29 (dd, $J$ = 8.7, 3.7 Hz, 1H), 3.92 (s, 3H), 3.28 (br s, 1H), 3.03–2.80 (m, 2H), 2.21 (s, 3H). $^{13}$C NMR (75 MHz, CDCl$_3$) $\delta$ 209.28 (s), 157.87 (s), 137.94 (s), 134.24 (s), 129.58 (s), 128.86 (s), 127.33 (s), 124.42 (s), 124.41 (s), 119.18 (s), 105.78 (s), 70.11 (s), 55.44 (s), 52.07 (s), 30.96 (s).

### Synthesis of MBH product 2-(hydroxy(4-nitrophenyl)methyl)cyclohex-2-en-1-one (5)

The product standard for the MBH reaction was synthesized as described previously[4]. The product was purified by flash column chromatography (9:1 pentane:ethyl acetate), yielding white crystals (190 mg, 7.7% yield). The spectral data were in accordance with the literature[68] (Supplementary Figs. 21 and 22). $^1$H NMR (300 MHz, CDCl$_3$) $\delta$ 8.18 (d, $J$ = 8.8 Hz, 2H), 7.54 (d, $J$ = 8.2 Hz, 2H), 6.82 (t, $J$ = 4.2 Hz, 1H), 5.60 (d, $J$ = 6.0 Hz, 1H), 3.57 (d, $J$ = 6.0 Hz, 1H), 2.49–2.39 (m, 4H), 2.00 (p, $J$ = 6.2 Hz, 2H). $^{13}$C NMR (75 MHz, CDCl$_3$) $\delta$ 200.26, 149.42, 148.33, 147.35, 140.30, 127.26, 123.66, 72.17, 38.55, 25.91, 22.50.

## Reporting summary

Further information on research design is available in the Nature Portfolio Reporting Summary linked to this article.

## Data availability

Input catalytic arrays as well as final design models have been deposited to *Zenodo* (https://doi.org/10.5281/zenodo.15979364 (ref. 69) and https://doi.org/10.5281/zenodo.15494858 (ref. 70), respectively). Crystal structures have been deposited in the PDB and are available via accession codes 9GBT, 9FW5, 9FW7, 9FWA, 9QDP and 9R7F.

## Code availability

The code for Riff-Diff, alongside detailed instructions for use, can be found at https://github.com/mabr3112/riff_diff_protflow.

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

**Acknowledgements** We thank A. Winkler and P. Macheroux for discussions about mass spectrometry and enzyme kinetics; and A. Green for providing the coordinates of transition state 4 of MBH. Additionally, we acknowledge the European Synchrotron Radiation Facility for provision of synchrotron radiation facilities, and thank the staff of the ESRF and EMBL Grenoble for assistance and support in using beamlines MASSIF-3, ID30B, and BM29. For open access, the authors have applied a CC BY public copyright license to any Author Accepted Manuscript version arising from this submission. M.B. and M.G.T. were supported by the Austrian Science Fund (FWF) grant 10.55776/DOC130 and trained within the framework of the PhD programme Biomolecular Structures and Interactions (BioMolStruct). M.H. thanks the University of Graz for financial support. W.E. was supported by the Austrian Science Fund (FWF) grant 10.55776/P30826. A.T., M.C., D.S. and G.O. were supported by funding from the European Research Council through an ERC Starting Grant (HelixMold 802217) and a FETOPEN project (ARTIBLED, 863170). This research was funded in whole, or in part, by the Austrian Science Fund (FWF) (10.55776/P30826 to G.O.).

**Author contributions** Conceptualization: M.B., A.T. and G.O. Methodology: M.B., A.T. and G.O. Software: M.B. and A.T. Validation: M.B., A.T., G.O., W.E. and C.F. Formal analysis: M.B., A.T., S.K. and W.E. Investigation: M.B., A.T., M.C., S.K., M.G.T., D.S., A.B., W.E., S.Y.H., M.A., C.F., M.M., H.L., F.R., M.H. and G.O. Resources: M.H. and G.O.; Data curation: M.B., A.T., A.B. and G.O. Writing: M.B., A.T. and G.O. Visualization: M.B., A.T. and S.K. Supervision: M.H. and G.O. Project administration: G.O. Funding acquisition: G.O. All authors read and contributed to the manuscript.

**Funding** Open access funding provided by Graz University of Technology.

**Competing interests** The authors declare no competing interests.

**Additional information**
**Correspondence and requests for materials** should be addressed to Gustav Oberdorfer.

**Extended Data Table 1 | Crystallization conditions and diffraction parameters for RAD and MBH designs**

| | RAD13 | RAD17 | RAD32 | RAD36 | MBH2 | MBH48 |
|---|---|---|---|---|---|---|
| **Data collecton** | | | | | | |
| Space group | P 1 21 1 | I 41 | P 31 | P 21 21 21 | C 1 2 1 | P 63 |
| Cell dimensions | | | | | | |
| $a$, $b$, $c$ (Å) | 38.04, 76.94, 59.62 | 107.11, 107.11, 52.42 | 60.51, 60.51, 41.32 | 46.60, 55.93, 73.34 | 56.78, 51.88, 72.61 | 76.06, 76.06, 59.64 |
| $\alpha$, $\beta$, $\gamma$ (°) | 90, 91.23, 90 | 90, 90, 90 | 90, 90, 120 | 90, 90, 90 | 90, 112.27, 90 | 90, 90, 120 |
| Resolution (Å) * | 59.61-2.43 (2.62-2.43) | 47.08-2.9 (3.004-2.9) | 32.45-2.0 (2.2-2.0) | 36.67-1.73 (1.792-1.73) | 36.75-1.13 (1.15-1.13) | 44.21-1.93 (2.08-1.93) |
| $R_{merge}$ * | 0.221 (2.641) | 0.111 (2.692) | 0.272 (2.989) | 0.1071 (1.316) | 0.05517 (0.8954) | 0.145 (2.226) |
| $I$ / $\sigma I$ * | 4.54 (0.47) | 11.89 (0.89) | 5.7 (0.72) | 15.26 (2.25) | 9.83 (1.12) | 13 (0.65) |
| Completeness (%) * | 98.94 (98.88) | 99.55 (98.77) | 99.89 (99.97) | 99.86 (100) | 96.77 (93.17) | 98.24 (91.49) |
| Redundancy * | 7.2 (6.9) | 10.4 (11) | 10.8 (10.4) | 13.3 (14.1) | 3.1 (1.9) | 20.7 (21) |
| **Refinement** | | | | | | |
| Resolution (Å) | 59.61-2.43 | 53.55-2.9 | 32.45-2.0 | 36.67-1.73 | 36.75-1.13 | 44.21 - 1.93 |
| No. Reflections | 107876 | 69818 | 164833 | 273610 | 494274 | 308129 |
| $R_{work}$ / $R_{free}$ | 0.2236 / 0.267 | 0.2349 / 0.2702 | 0.2162 / 0.2717 | 0.1903 / 0.2341 | 0.1389 / 0.1675 | 0.2011 / 0.2514 |
| No. atoms | | | | | | |
| Protein | 3134 | 1477 | 1503 | 1528 | 1601 | 1497 |
| Ligand/ion | 47 | 22 | 0 | 2 | 15 | 64 |
| Water | 25 | 0 | 39 | 180 | 420 | 88 |
| $B$-factors | | | | | | |
| Protein | 66.76 | 67.82 | 33.25 | 22.86 | 14.71 | 51.58 |
| Ligand/ion | 73.2 | 29.35 | | 28.51 | 37.57 | 68.58 |
| Water | 58.42 | | 36.26 | 34.08 | 34.07 | 53.16 |
| R.m.s. deviations | | | | | | |
| bond lengths (Å) | 0.003 | 0.011 | 0.002 | 0.007 | 0.003 | 0.006 |
| Bond angles (°) | 0.53 | 1.41 | 0.53 | 0.73 | 0.7 | 0.78 |

*Values in parenthesis are for highest-resolution shell

# Reporting Summary

## Statistics

For all statistical analyses, confirm that the following items are present in the figure legend, table legend, main text, or Methods section.

| n/a | Confirmed | |
|---|---|---|
| ☒ | ☐ | The exact sample size (*n*) for each experimental group/condition, given as a discrete number and unit of measurement |
| ☐ | ☒ | A statement on whether measurements were taken from distinct samples or whether the same sample was measured repeatedly |
| ☐ | ☒ | The statistical test(s) used AND whether they are one- or two-sided<br>*Only common tests should be described solely by name; describe more complex techniques in the Methods section.* |
| ☒ | ☐ | A description of all covariates tested |
| ☒ | ☐ | A description of any assumptions or corrections, such as tests of normality and adjustment for multiple comparisons |
| ☐ | ☒ | A full description of the statistical parameters including central tendency (e.g. means) or other basic estimates (e.g. regression coefficient) AND variation (e.g. standard deviation) or associated estimates of uncertainty (e.g. confidence intervals) |
| ☐ | ☒ | For null hypothesis testing, the test statistic (e.g. *F*, *t*, *r*) with confidence intervals, effect sizes, degrees of freedom and *P* value noted<br>*Give P values as exact values whenever suitable.* |
| ☒ | ☐ | For Bayesian analysis, information on the choice of priors and Markov chain Monte Carlo settings |
| ☒ | ☐ | For hierarchical and complex designs, identification of the appropriate level for tests and full reporting of outcomes |
| ☒ | ☐ | Estimates of effect sizes (e.g. Cohen's *d*, Pearson's *r*), indicating how they were calculated |

*Our web collection on statistics for biologists contains articles on many of the points above.*

## Software and code

Policy information about availability of computer code

| Data collection | https://github.com/mabr3112/riff_diff_original, https://github.com/mabr3112/riff_diff_protflow |
|---|---|
| Data analysis | All data analysis was performed with python scripts. These scripts are available from the link above. |

For manuscripts utilizing custom algorithms or software that are central to the research but not yet described in published literature, software must be made available to editors and reviewers. We strongly encourage code deposition in a community repository (e.g. GitHub). See the Nature Portfolio guidelines for submitting code & software for further information.

## Data

Policy information about availability of data

All manuscripts must include a data availability statement. This statement should provide the following information, where applicable:
- Accession codes, unique identifiers, or web links for publicly available datasets
- A description of any restrictions on data availability
- For clinical datasets or third party data, please ensure that the statement adheres to our policy

The datasets generated and analyzed during the current study are available at https://doi.org/10.5281/zenodo.15494922 and from the corresponding author on request. An updated and easier to use version of Riff-Diff is available at https://github.com/mabr3112/riff_diff_protflow. Diffraction data and corresponding atomic models for the designs have been deposited in the Protein Data Bank under accession codes 9GBT, 9FW5, 9FW7, 9FWA , 9R7F and 9QDP respectively.

# Research involving human participants, their data, or biological material

Policy information about studies with human participants or human data. See also policy information about sex, gender (identity/presentation), and sexual orientation and race, ethnicity and racism.

| | |
|---|---|
| Reporting on sex and gender | *Use the terms sex (biological attribute) and gender (shaped by social and cultural circumstances) carefully in order to avoid confusing both terms. Indicate if findings apply to only one sex or gender; describe whether sex and gender were considered in study design; whether sex and/or gender was determined based on self-reporting or assigned and methods used.*<br>*Provide in the source data disaggregated sex and gender data, where this information has been collected, and if consent has been obtained for sharing of individual-level data; provide overall numbers in this Reporting Summary. Please state if this information has not been collected.*<br>*Report sex- and gender-based analyses where performed, justify reasons for lack of sex- and gender-based analysis.* |
| Reporting on race, ethnicity, or other socially relevant groupings | *Please specify the socially constructed or socially relevant categorization variable(s) used in your manuscript and explain why they were used. Please note that such variables should not be used as proxies for other socially constructed/relevant variables (for example, race or ethnicity should not be used as a proxy for socioeconomic status).*<br>*Provide clear definitions of the relevant terms used, how they were provided (by the participants/respondents, the researchers, or third parties), and the method(s) used to classify people into the different categories (e.g. self-report, census or administrative data, social media data, etc.)*<br>*Please provide details about how you controlled for confounding variables in your analyses.* |
| Population characteristics | *Describe the covariate-relevant population characteristics of the human research participants (e.g. age, genotypic information, past and current diagnosis and treatment categories). If you filled out the behavioural & social sciences study design questions and have nothing to add here, write "See above."* |
| Recruitment | *Describe how participants were recruited. Outline any potential self-selection bias or other biases that may be present and how these are likely to impact results.* |
| Ethics oversight | *Identify the organization(s) that approved the study protocol.* |

Note that full information on the approval of the study protocol must also be provided in the manuscript.

# Field-specific reporting

Please select the one below that is the best fit for your research. If you are not sure, read the appropriate sections before making your selection.

☒ Life sciences  ☐ Behavioural & social sciences  ☐ Ecological, evolutionary & environmental sciences

For a reference copy of the document with all sections, see nature.com/documents/nr-reporting-summary-flat.pdf

# Life sciences study design

All studies must disclose on these points even when the disclosure is negative.

| | |
|---|---|
| Sample size | *Describe how sample size was determined, detailing any statistical methods used to predetermine sample size OR if no sample-size calculation was performed, describe how sample sizes were chosen and provide a rationale for why these sample sizes are sufficient.* |
| Data exclusions | *Describe any data exclusions. If no data were excluded from the analyses, state so OR if data were excluded, describe the exclusions and the rationale behind them, indicating whether exclusion criteria were pre-established.* |
| Replication | Kinetic measurements, CD spectroscopy, protein expression performed in triplicate. |
| Randomization | *Describe how samples/organisms/participants were allocated into experimental groups. If allocation was not random, describe how covariates were controlled OR if this is not relevant to your study, explain why.* |
| Blinding | *Describe whether the investigators were blinded to group allocation during data collection and/or analysis. If blinding was not possible, describe why OR explain why blinding was not relevant to your study.* |

# Reporting for specific materials, systems and methods

We require information from authors about some types of materials, experimental systems and methods used in many studies. Here, indicate whether each material, system or method listed is relevant to your study. If you are not sure if a list item applies to your research, read the appropriate section before selecting a response.

## Materials & experimental systems

| n/a | Involved in the study |
|-----|----------------------|
| ☒ ☐ | Antibodies |
| ☒ ☐ | Eukaryotic cell lines |
| ☒ ☐ | Palaeontology and archaeology |
| ☒ ☐ | Animals and other organisms |
| ☒ ☐ | Clinical data |
| ☒ ☐ | Dual use research of concern |
| ☒ ☐ | Plants |

## Methods

| n/a | Involved in the study |
|-----|----------------------|
| ☒ ☐ | ChIP-seq |
| ☒ ☐ | Flow cytometry |
| ☒ ☐ | MRI-based neuroimaging |

## Plants

| | |
|---|---|
| Seed stocks | *Report on the source of all seed stocks or other plant material used. If applicable, state the seed stock centre and catalogue number. If plant specimens were collected from the field, describe the collection location, date and sampling procedures.* |
| Novel plant genotypes | *Describe the methods by which all novel plant genotypes were produced. This includes those generated by transgenic approaches, gene editing, chemical/radiation-based mutagenesis and hybridization. For transgenic lines, describe the transformation method, the number of independent lines analyzed and the generation upon which experiments were performed. For gene-edited lines, describe the editor used, the endogenous sequence targeted for editing, the targeting guide RNA sequence (if applicable) and how the editor was applied.* |
| Authentication | *Describe any authentication procedures for each seed stock used or novel genotype generated. Describe any experiments used to assess the effect of a mutation and, where applicable, how potential secondary effects (e.g. second site T-DNA insertions, mosiacism, off-target gene editing) were examined.* |

