## [Peer Review File · Nature]

Computational enzyme design by catalytic motif scaffolding

Corresponding Author: Professor Gustav Oberdorfer

Version 0:

Reviewer comments:

Referee #1

(Remarks to the Author)

In the research paper "Computational Design of Highly Active De Novo Enzymes," Oberdorfer et al. introduce Riff-Diff, a method for de novo enzyme design that transplants catalytic arrays into newly generated protein backbones. The method incorporates artificial motif libraries by embedding catalytic arrays into alpha-helical fragments, which are then scaffolded using RFdiffusion. To improve binding pocket formation, a custom auxiliary potential and placeholder alpha-helices were used, leading to optimized pockets without relying on substrate auxiliary potentials.

Riff-Diff's iterative refinement cycles further optimize backbone-side chain interactions, guided by Rosetta's FastDesign protocol and ProteinMPNN for sequence design. The method was applied to retro-aldoases, successfully scaffolding a catalytic tetrad, resulting in designs with catalytic rates exceeding five-millionfold acceleration over the uncatalyzed reaction.

Of the 35 successfully cloned designs, all led to soluble proteins, with many showing measurable catalytic activity. Structural and biochemical validation, including SAXS, CD spectroscopy, and mass spectrometry, confirmed structural integrity and thermal stability of the designs.

Overall, this study presents an application of established computational design tools to scaffold catalytic arrays within de novo protein backbones. The authors employed key existing methodologies such as RFdiffusion, ProteinMPNN, and Rosetta FastDesign, combining them effectively to create a small validation library of retro-aldol enzymes. The research also includes important biochemical validation steps to assess the activity and structural stability of the designed enzymes.

The work demonstrates skillful use of cutting-edge protein design tools in a context that shows potential, though it is important to note that the core methods (i.e., RFdiffusion, ProteinMPNN, and Rosetta FastDesign) are not novel. Furthermore, the claim of generating "highly active enzymes" is not entirely supported, as there is no transparent comparison with state-of-the-art de novo enzymes or artificially evolved variants, making it challenging to fully evaluate the method's impact. Additionally, the approach is limited to cases where known catalytic motifs are available, restricting its generalizability for novel enzymatic functions.

Despite these limitations, this work holds relevance for researchers in enzyme design and biocatalysis but I feel it may be better suited for a specialized journal like Nature Catalysis after addressing the concerns outlined below.

Novelty of the study

The novelty of this study is limited by its heavy reliance on previously established tools and methods, such as RFdiffusion, ProteinMPNN, and Rosetta FastDesign. These technologies have already demonstrated success in key areas of protein design, including scaffolding catalytic sites, designing protein monomers, binders, and symmetric assemblies. The RFdiffusion model has been extensively validated for a broad range of design challenges, such as topology-constrained design, metal-binding protein creation, and enzyme active site scaffolding. As a result, while the study claims to achieve high activity in de novo enzymes, it primarily refines existing methods without introducing fundamentally new concepts or frameworks for protein design.

Comments on the manuscript

General:

1. A direct comparison with other de novo designed retro-aldolases is essential and should be included to establish a baseline for evaluating the success of this study. Additionally, the authors need to substantiate their claim that their "highly active de novo enzyme" rivals those optimized by in vitro evolution by providing a direct comparison of catalytic performance under the same experimental conditions (K_m , k_{cat} , k_{cat}/K_m , TTN etc). In this respect, the captions for Figs 1 and 2 contain strong claims that are not well supported by the presented data and illustrations. For instance, the caption for Fig. 2 states, "Activity of designed retro-aldol enzymes far exceeded those of previous zero-shot designs." However, the figure does not clearly convey this observation. I suggest that the authors include a table and a bar plot that explicitly presents the k_{cat} and K_m values for: a) the best-performing de novo designs from the literature, b) the best-performing engineered variants from the literature, and c) the RiffDiff variants.

2. Do the authors anticipate that their method can be applied in cases where no functional catalytic arrays are known?

Specific:

1. Page 2, line 11: I do not agree that certain reactions are completely unsuitable for high-throughput screening. While it is valid to acknowledge that time and resources can sometimes be limited, or that the product may not justify the screening effort, in principle, every reaction could be screened at a higher throughput. This ultimately depends on the effort invested in developing analytical methods. Further, the authors do not mention numerous computational strategies other than de novo design have been developed to reduce screening effort.

<https://www.sciencedirect.com/science/article/pii/S0959440X21000154>

<https://www.nature.com/articles/s41592-019-0496-6>

<https://www.nature.com/articles/s41589-024-01712-3>

2. Page 2, lines 35-37: Next to retro aldolases, decades of efforts on de novo design and optimization of Kemp Eliminase should be acknowledged in a sentence providing suitable citations. Especially because Kemp is later used as illustrative example for discriminating between active and inactive designs using MD simulation (line 45 and following).

<https://www.sciencedirect.com/science/article/pii/S0022283611000842>

<https://www.nature.com/articles/nature12623>

<https://www.nature.com/articles/s41467-020-18619-x>

3. Page 3, line 1: Next to the relevance of conformational preorganization on protein level, MD simulations studying enzyme substrate interactions have been demonstrated to provide relevant insights in the context of optimized de novo enzymes.

<https://pubs.acs.org/doi/full/10.1021/acs.jcim.3c00002>

Thus, I recommend that the authors also study their designs using MD simulations of enzyme-substrate complexes rather than solely focusing on protein level preorganization.

4. Page 4, lines 4-8: It is not fully clear why, where, and how the authors added the additional alpha-helix. How did they decide on length and geometric properties of this helix? In general, how did you select the sequence and structures of the helical fragments?

5. Page 4, line 24: Please explain why you used the underperforming ESMFold instead of AlphaFold2 in this step – tradeoff between speed and accuracy? Later (line 29) you use AlphaFold2.

6. Page 5, lines 11-12: Please specify how you analyzed the buried active site and how you defined which binding site is buried and which not.

7. Page 5, lines 11-12: Please specify what you mean with "most" (e.g. 20/35) and how "measurable" was defined.

8. Page 5, line 15: I understand that structural similarity (TM-score) is the most relevant metric. In addition, it would be interesting how similar the closest natural sequences are (sequence identity, sequence similarity, coverage).

9. Page 6, lines 1-2: I encourage the authors to also determine protein melting temperatures (T_m) as a measure for stability.

10. Page 7, line 4: Please explain how you defined "partially functional".

11. Page 7, line 24: It is very common that there is a high deviation between MD simulation replicates, for which reason several replicates should be performed (at least 5, better 10-20). In the context of Kemp eliminase it has been shown that many short simulations can help to achieve the necessary throughput if computational resources are limited. Please see:

<https://pubs.acs.org/doi/full/10.1021/acs.jcim.3c00002>

12. Page 10, lines 9-10: Please specify what you used as positive and negative control.
13. Page 10, line 13: I suggest performing triplicates instead of duplicates.
14. Page 10, line 41: Why did you switch from 50 to 100 mg/L kanamycin?
15. Page 11, lines 3-4: DNase & lysozyme concentrations are missing.
16. Page 11, line 9: Do you mean $1/16 = 0.062$ mg?
17. Page 11, line 12: Please specify the ingredients of the storage buffer.
18. Page 11, line 16: Circular Dichroism (CD) and thermal denaturation
19. Page 12, line 7: Please specify what you used as precipitant solution.
20. Page 12, line 30: Please specify the full range and step size of the serial dilution.
21. Page 13, line 2: In my opinion an introductory sentence specifying Gromacs version and purpose of the experiment is necessary.
22. Page 13, lines 12-13: NVT and NPT equilibration seem very short for de novo designed structures. Please provide quality measures of the equilibration success (RMSD, radius of gyration, potential energy, temperature, and pressure). It's common to apply positional restraints to the heavy atoms of the backbone or specific regions of the protein during the initial stages of equilibration to prevent large conformational changes while the solvent is equilibrating. If restraints were used, specifying the force constant and the atoms to which they were applied would be beneficial.
23. Page 13, line 18: Using the LINCS algorithm for bond constraints is standard practice, particularly for systems involving hydrogen atoms. However, clarifying which bonds are constrained (e.g., all bonds involving hydrogen) is necessary.

Comment on statistics

The authors provide information on the number of independent experiments conducted as well as standard deviation values.

Comment on the PDB validation reports

Please doublecheck the clashes, the number of ligand and solvent molecules and ensure that all units (e.g. for angles and distances) are specified in the SI.

(Remarks on code availability)

I thank the authors for providing access to their GitHub repository. However, I have some concerns in terms of license, usability, and reproducibility.

- 1) The license under which the code can be used by others should be specified (e.g. MIT).
- 2) Documentation on a) how to install dependencies, b) needed hardware and software requirements, c) installation and primary citation of external tools, and d) workflow and how to use the provided code is completely missing.
- 3) I strongly recommend providing some kind of workflow script or jupyter notebook; the repository contains more than 10 separate Python scripts without any documentation on how and in which order to use them.
- 4) I strongly recommend providing a Docker container or at least conda yml files with the needed software.

Given that the primary contribution of the authors' work is a novel method for the de novo design of efficient enzymes, it would be highly beneficial to provide a comprehensive, reproducible workflow or tool to facilitate broader use. This is essential for the technology's adoption and for maximizing its impact on future research. While I recognize the expertise shown in utilizing external software and tools, the innovative value of RiffDiff as a new technology or methodology would be significantly enhanced if it were made usable for other researchers.

Referee #2

(Remarks to the Author)

In this article, Oberdorfer and colleagues introduce RiffDiff, a de novo enzyme design pipeline that constructs enzymes for specific reactions by building an artificial protein scaffold around a designated catalytic motif with its associated ligand. RiffDiff begins with the side chain coordinates of a preexisting catalytic motif extracted from the PDB—here, the catalytic tetrad of evolved retro-aldolase RA95.5-8F—embedding each tetrad residue on its own alpha-helical peptide to determine compatible positions of backbone atoms that avoid steric clashes while maintaining the catalytically productive arrangement

of functional groups. The method then fills backbone gaps between these peptides using RFDiffusion, designs sequences to stabilize the scaffold using ProteinMPNN, and refines active-site rotamer configurations using the FastDesign and CoupledMoves protocols implemented in the Rosetta protein design suite.

Using RiffDiff, the authors designed 36 retro-aldolases, 30 of which displayed activity. The most active designs are the most efficient de novo retro-aldolases reported to date, though their catalytic rates remain modest ($k_{cat} \leq 0.036 \text{ s}^{-1}$). Crystal structures for four designs confirm the accuracy of the designed active-site configurations and/or overall folds. The authors conclude the article with a discussion offering insights and suggestions for further refinement. This study demonstrates impressive results, and several aspects of RiffDiff's methodology, such as its use of an artificial motif library and placeholder alpha-helices for entry-channel design, are particularly innovative. RiffDiff promises to become a valuable tool in de novo enzyme design, complementing the traditional theozyme-based approach. I recommend publication of this article following revisions to clarify methods, temper unsupported claims, and add important missing data.

1. Overstatements and Misleading Claims

> The title of this article is misleading. It implies high activity even though these enzymes are not highly active despite being the most active computationally designed retro-aldolases to date. For example, the most active design, RA29, has a catalytic efficiency of $290 \text{ M}^{-1} \text{ s}^{-1}$, which is orders of magnitude lower than the average natural enzyme (k_{cat}/K_M of $\sim 100,000 \text{ M}^{-1} \text{ s}^{-1}$) or even the most active artificial retro-aldolase (RA95.5-8F, $k_{cat}/K_M = 34,000 \text{ M}^{-1} \text{ s}^{-1}$). This value is instead comparable to those of other de novo enzymes, such as Kemp eliminases HG3 ($k_{cat}/K_M = 1300 \text{ M}^{-1} \text{ s}^{-1}$, see Blomberg et al., Nature 2013) and KE59 ($k_{cat}/K_M = 160 \text{ M}^{-1} \text{ s}^{-1}$; see Khersonsky et al., PNAS, 2012). Furthermore, k_{cat} values are modest, with a maximum of 0.036 s^{-1} . These results indicate that RiffDiff produces de novo enzymes with modest activity, contradicting the title's claim that the enzymes are "highly active". Thus, the title should be revised.

> The key achievement of this paper is not that the enzymes' activity is high, but that de novo enzymes can be constructed by building a completely new protein to scaffold a catalytic motif and form a binding pocket for the ligand—without relying on existing protein scaffolds. This point, rather than overstating activity, should be reflected in the title.

> The abstract claims that the "designs exhibit a high fold diversity". Yet, all designs are single-domain, alpha-helical proteins, which doesn't suggest high diversity. The authors likely mean that the designs adopt distinct folds; this should be clarified to avoid confusion.

> The claim of "proficient retro-aldol enzymes" in the abstract is misleading. Although these enzymes are the most active computationally designed retro-aldolases to date, they are not proficient biocatalysts (as discussed above).

2. Preorganization Misinterpretation

> The authors use MD to evaluate catalytic residue preorganization but MD, as applied here, cannot measure this. Preorganization refers to active-site groups not having to undergo significant rotation to stabilize the transition state, since they are already pointing in the correct direction (see Jindal & Warshel, Proteins, 2017). This can only be shown by comparing active-site structures with and without a transition state (or analogue). Here, unbound structures were analyzed, so preorganization cannot be determined, as conformational changes may occur upon transition-state formation. The authors instead have assessed rigidity of the catalytic tetrad, indicating its ability to adopt productive or unproductive conformations, given the assumption that the designed configuration is catalytically productive. The authors should thus remove all discussion of preorganization from the manuscript and focus on rigidity/flexibility and/or productive/unproductive conformations.

> Along the same lines, on Fig. S9, it is stated that the catalytic tetrad is well preorganized. Notwithstanding the inability of MD as performed here to assess preorganization, the Y180 residue is shown to be highly flexible, which presumably would not be ideal for efficient catalysis.

3. Conclusions not fully supported by data

> The statement "High activity originated from catalytic tetrad" (section title, p. 6) is not fully supported by the data. While the authors mutated Lys to Ala to demonstrate the importance of the designed nucleophile, they did not mutate the other tetrad residues to confirm their individual contributions to catalysis. Without this data, it is unclear whether designing the entire tetrad was necessary or if a subset of these residues would suffice. To substantiate their claims regarding the catalytic tetrad's role, the authors must create knockouts for each tetrad residue in a subset of their enzymes and report the resulting k_{cat} and K_M values (as done by Obexer et al., Nat. Chem., 2017).

> On p. 7, the authors state that the preference of "RAD29 to form (R)-methanol in the forward reaction (aldol addition) further supports the successful design of a specific substrate binding mode and participation of all tetrad residues". This is incorrect: the reported 60% enantiomeric excess indicates an 80:20 R:S ratio, demonstrating that the active site is not specific to a single substrate binding mode. This is especially evident when compared to RA95.5-8F, which achieved an enantiomeric excess of 99.2:0.8 (Obexer et al., Nat. Chem., 2017). Furthermore, this result does not prove the participation of all tetrad residues in catalysis.

4. Methods Section Completeness

> The methods section lacks essential details on the computational design procedure, which is necessary for reproducibility. Missing specifics include: probabilities of catalytic tetrad rotamers used to generate the artificial motif library (and associated phi/psi bonds), details for energy calculations, residue identities allowed at active-site positions during FastDesign/CoupledMoves (i.e. the searched sequence space), definitions of ligand interactions (e.g., geometric definition of contacts), amino acid probability cutoffs used to select sequence space for active-site optimization, inhibitor attributes (including treatment of covalent bond with Lys), design filtering criteria (e.g., predicted side-chain RMSDs), etc. Authors must include a detailed description of RiffDiff in the methods section and provide scripts or code for using it. The article should not be published without these.

> Page 4, line 29 mentions that final sequences were evaluated with AlphaFold2 and ranked using metrics; these metrics

should be specified.

> Please clarify the term "free folding energies"; does this refer to a computed potential energy difference between the folded structure and an unfolded one? Or is this a true free energy that includes entropy? How is this calculated?

> In general, the methods section as a whole should be revised to enhance clarity and provide sufficient details for reproducibility, including descriptions of negative/positive controls, plasmids, promoters, enzyme units used for kinetic assays, extinction coefficients for concentration calculations, chemical purity and source, etc.

5. Data Presentation

The most important data in this article are the kinetic parameters (kcat, KM) and thermodynamic stabilities of the designed enzymes, which are currently buried in supplementary materials. Please consolidate these data into a single table in the main text and convert Cm values to ΔG to provide more meaningful insights.

6. Missing pKa Values

Please include pKa values for the catalytic lysine, at least for the most active designs (RA29/RA35) and those with available crystal structures (RAD13/RAD17/RAD32/RAD36). The discussion of catalytic tetrad activity is incomplete without these, since low kcat could in part be due to high pKa.

7. Clarifications needed

> What do the authors mean by "zero shot"? From what I understand, zero shot is used to describe a machine learning model's ability to perform a task it hasn't been explicitly trained on. Yet, the ML methods used in RiffDiff (i.e. RFDiffusion and ProteinMPNN) were specifically trained to design protein structures and sequences, which is what they're used to do here. It is important to note that in this article, catalysis is not explicitly designed, since there is no QM calculation used to predict activity. Instead, catalysis is inferred from the sequence and structure of the active site, making it unclear why the term zero-shot is used here.

> The term "catalytically potent theozyme" is also unclear. Is this potency based on theozyme DFT energy? Likely, the authors mean that the catalytic tetrad motif was empirically determined to be more efficient for the retro-aldol reaction than simpler motifs, such as the Lys/Glu pair used by Althoff et al. (Prot. Sci., 2012) to design RA95. Additionally, the tetrad with bound inhibitor used here is not a theozyme, as it is not a QM model of catalytic groups stabilizing a transition state. I recommend that the authors verify whether they are properly using the term theozyme throughout their article (see Tantillo et al., Curr. Opin. Chem. Biol., 1998).

> The authors analyze crystal structures to explain activity differences between variants. They focus on analyzing the accuracy of the catalytic tetrad coordinates between design model and crystal structure. Yet, they do not discuss whether the binding pocket itself is amenable to efficient binding of the substrate. From looking at the crystal structures, I can see that not all of them have a well-defined binding pocket for the substrate. In the absence of crystal structures with a bound substrate analogue, the authors should perform docking to verify whether the substrate binding pocket can make interactions with this ligand as designed. It may also help to explain the reported ee.

> The enantiomeric excess of 60% obtained for synthesis of methodol by RAD29 indicates an 80:20 preference for the R-enantiomer, which suggests that the aldehyde substrate binds in multiple orientations within the active site. This result challenges the notion that a binding pocket complementary to the substrate has been designed. Given that the design procedure used a placeholder helix during RFDiffusion, is it possible that removal of this helix created a pocket that was not ideal for the substrate? The authors should discuss this potential limitation of the method.

8. Missing References

> Page 2, line 20: "minimal active sites, called theozymes, can be constructed by placing amino acid functional groups around a computational model of a transition state in a stabilizing geometry." The reference cited here (Zanghellini et al., Prot. Sci., 2006) is not appropriate, as it describes the RosettaMatch protocol instead of the concept of a theozyme. You should instead cite the work of Tantillo & Houk on defining theozymes (Curr. Opin. Chem. Biol., 1998).

> Page 3, line 4: "Despite all these findings, recent de novo enzyme design strategies rarely optimized preorganization explicitly." There is a recent example (Rakotoharisoa et al., JACS, 2024) where preorganization was optimized during the enzyme design procedure and experimentally validated using crystal structures with and without a transition-state analogue. This work should be cited.

9. Missing data

> Please provide amino acid sequences of all 36 designs in a FASTA format.

> Please provide PDB files for all design models, with inhibitor bound (since these were designed with Rosetta).

> Please include a figure that shows the electron density of catalytic residues, to confirm that these were properly modelled. Inspection of the RA36 structure suggests that the tetrad Asn and Lys residues could benefit from improved modeling.

10. Minor comments

> p. 3, line 42: "high specificity" should be replaced by "narrow specificity". High specificity refers to a high kcat/KM value, whereas narrow specificity is used to describe an enzyme's ability to react with only a small number of substrates.

> p. 4: "vanilla RFDiffusion" needs clarification—is this a specific version? If so, specify version or release date.

> p. 5, line 10: Name the ligand involved in binding-pocket interactions.

> Avoid using "exceptional" to describe model accuracy, as this is subjective. Values of 0.89 Å and 1.2 Å RMSD, while accurate, are not "exceptional."

> Typos: replace "lysins" with "lysines"; p. 9, line 1: "catalytic site" should be "catalytic motif."

> Fig. S6: Confirm if this shows racemic methodol.

> Indicate error and number of technical/biological replicates for all figures.

> Table S4's title reads like a footnote, please fix.

- > Fig. 1c, middle graph: Improve line visibility.
- > Fig. 1e: Increase size for readability.
- > Fig. 2f: Add RA95 for comparison.
- > Fig. 2: Clarify the linear regression model based on AF2 prediction confidence.
- > Rework Fig. 3 legend to address issues regarding incorrect use of terms like preorganization, exceptional, and theozyme.

(Remarks on code availability)

Code was provided too late for testing (review was almost complete). It would help if code was provided at the same time as manuscript. Same for crystal structures.

Referee #3

(Remarks to the Author)

In the manuscript at hand titled "Computational design of highly active de novo enzymes" Braun et al. present a strategy for computationally predicting enzymes using a newly developed machine-learning-based prediction algorithm called "Riff-Diff". The authors claim that this new strategy can be in principle applied to create enzymes catalyzing various chemical reactions given the transition state of the reaction and an amino acid constellation that stabilizes that transition state. In the manuscript at hand, the authors focus on designing a (retro-)aldolase as a proof-of-principle. The designs resulting from their new design pipeline impressively generate catalytically active de novo enzymes within few numbers of designs (35 tested) and without the need for further directed evolution campaigns ("zero-shot").

The Riff-diff design pipeline starts with a theozyme or catalytic array (a minimal set of amino acid residues that stabilize the transition state of the reaction) as an input. The Riff-Diff algorithm then aims to construct a protein backbone from helical peptide fragments which already place the catalytically relevant amino acid residues precisely in the catalytically active geometry (termed preorganization). A key step in the pipeline is the generation of an "artificial motif library" by combining rotamer-varied fragments into "motifs", selecting them for physically sensible motifs in terms of compatibility of the individual rotamers with their backbone and avoiding steric clashes between fragments, and finally ranking the motifs according to the probabilities of the rotamers involved. The selected motifs are then further modified with an additional helical placeholder-fragment, which serves to promote the formation of a tighter binding pocket in the next step. Next, RFDiffusion with a modified auxiliary potential is used to construct the remaining backbone scaffold, while preserving the fragments, followed by Rosetta FastDesign-based optimization of residue-substrate interactions, and ranking with ESMFold. In an iterative manner the top scaffolds are re-subjected to additional cycles of backbone optimization. After multiple optimization cycles CoupledMoves further refines protein-substrate interaction, and ProteinMPNN finalizes the structure. The final structures are then ranked using AlphaFold2 prediction.

The entire design pipeline has been compiled into a single script, minimizing user input and thus potentially making it an easy/easier-to-use tool. Using their script, the authors proceeded to design de novo retro-aldolases targeting the compound methodol. Final structures showed a mostly high agreement with the targeted catalytic site geometry (observed differences between model and determined structures at or below 1 Å) and favourable predicted properties. When tested experimentally, some of the designed enzymes showed remarkably high catalytic performance. Two designs specifically, RAD35 and RAD29, approached 5·10⁶-fold reaction rate acceleration and the latter even rivalled a previously evolved retro-aldolase in catalytic efficiency. That these activities were achieved without further engineering is already an impressive and (to the best of the authors' and our knowledge) ground-breaking achievement.

To correlate protein structure, dynamics and activity, crystal structures of four designs were obtained, and MD simulations following the catalytic residues were performed. Here, discrepancies emerged that put the importance of high preorganization of catalytic residues and precise positioning into question.

While in principle the presented design pipeline can be applied to any reaction with a known transition state that can be stabilized by amino acid residues, this versatility was not shown within the manuscript itself. Furthermore, it has yet to be demonstrated that the remarkably high success rate for the enzyme designs do transfer to other reactions and catalytic mechanisms. The enzymatic model reaction used here, has already undergone extensive experimental optimization in previous works and the idealized transition state geometry is therefore well established. The impact of lesser optimized catalytic systems on the success rate would be an interesting property of this new design tool and should be addressed in future work.

Overall, the work seems to deliver a valuable high-precision enzyme design tool. Predicting de novo enzyme structures with high activity would immensely facilitate the development of biocatalytic processes by reducing the amount of screening and directed evolution campaigns to a minimum. The work therefore represent an important step in supporting the development of greener alternatives to conventional chemical transformation and is of moderately high impact or high impact if the aforementioned versatility is proven in a later work.

Below are additional major and minor comments that should be addressed:

Major points:

1. The manuscript did not include or link to the source code for the Riff-Diff combined script. This should be added in the final version of the manuscript as several questions and minor comments could likely be resolved with it.

2. Fig. 1d: The caption of this figure requires a more detailed description of the data shown. Additionally, either a legend should be added or different types of points should be explained in the caption (e.g. do the white square (presumably) represent the median or average? Do the horizontal lines represent the total range or something different?).

3. It is unfortunate that no crystal structure could be obtained for the two most promising designs, RAD29 and RAD35. The conclusions drawn on p. 7, lines 29ff on precise residue placement thus rely primarily on the comparison of only one highly active design (RAD32) with three no-/low activity designs. Furthermore, as the authors point out themselves, the geometry RAD32 deviates from the intended design as the tyrosine of the tetrad has been unintentionally replaced by another tyrosine at a different position (a circumstance that to a lesser extent reminds of a criticism the authors made themselves in an earlier passage on p. 2, lines 40ff). Therefore, the conclusions on the importance of precise residue placement are a little weak and speculative and should be treated as such. Further experimental and computational work would be needed to elucidate the underlying causes as the authors correctly mention. This, however, might be out of the scope of the manuscript.

4. The rationale to use a selection of rotamers for library building was to promote preorganization (p. 3, line 21). Yet, one of the final conclusions is that preorganization does not seem to be a critical factor for increased activity. Additionally, preorganization in the different designs is also only achieved to varying degrees. This unexpected finding raises the question which of the many considerations in the RiffDiff pipeline is critical for the high success rate or whether it is a complex interplay of all components. This warrants further discussion.

5. The kinetic characterization of the designed retro-aldolases were performed in technical triplicate according to the caption of Fig. S6. The kinetic characterization of at least those designs that are critical for the findings in the main text (i.e. RAD29, RAD35, RAD13, RAD17, RAD36, RAD32) should also be validated as a biological duplicate (e.g. new, independent protein expression batch). Furthermore, the kinetic parameters reported in Table S3 should include measures of uncertainty (standard deviation or confidence intervals).

Minor points

- A reaction scheme would be nice to include for the retro-aldol reaction of methodol.
- Page 4, lines 6-12: The placement of the auxiliary potential and placeholder helix is described rather cryptically.
- Page 4, lines 14-19: How many iterative cycles are used for scaffolding or is the number based on some criterium?
- Page 4, line 30: The phrasing that AlphaFold2 ranks using "a set of metrics" is again very vague and might be addressed.
- Page 4, lines 44ff: Ambiguous phrasing: pLDDT values correspond to individual residues and the average pLDDT corresponds to the pLDDT values over a given structure. The value(s) given by the authors, is then presumably the average of the whole-sequence averages? Please clarify phrasing and also already cross-reference the corresponding swarm plot in Fig. 1d.
- Page 5, lines 11ff: Why were 36 sequences chosen? How many designs had to be discarded due to the post-design criteria? How was the accessibility of the binding pocket assessed? The ambiguity in the text suggests it was through manual inspection. Please clarify.
- The designs chosen for experimental validation were likely also influenced on the final ranking of the design pipeline. Does that ranking correlate with activity or any other experimental or structural property?
- Page 5, line 20: For the Michaelis-Menten experiments mentioned in this line, were these separate assessments or the same described on p. 6, lines 18ff? If so, please clarify phrasing (e.g. cross-referencing the paragraph). Mentioning the Michaelis Menten experiments here implies that batch-purification of the proteins has already happened, which is somewhat confusing as the next paragraph starts with the assessment of the that very batch purification. Please consider rephrasing for improving clarity.
- Fig. 2a: Conditions/instrumentalization of SEC analysis is missing. Please add these in the methods section or in the supplementary information. Furthermore, How were monomer elution times determined/ how was the instrument calibrated? At least one of the proteins (RAD30) seems to elute later than the others. Was it significantly shorter/ smaller than the other constructs? Providing the sequences as pointed out in the major comments, might be helpful to understand this.
- Fig. 2b: The label of the y-axis is a little confusing. If the y-axis depicts the "loss" of the 220 nm signal, the axis labels should start at 0, not 100% (in other words, there is no loss without heat treatment).
- Fig. 2c and p. 5, line 30: The text states that proteins were produced in high yields, implying that this is true for the majority of the designs. However, the majority of protein yields is below 20 mg per litre of culture, which can be considered moderate yield, with three designs having good yields of ~30 mg/L and above.
- Fig. 2c: An additional legend or explanation in the caption is needed with regards to different elements in the figure (see also major comment for Fig. 1d). Additionally, is there a reason for RAD5 to be such an outlier?
- Please provide more detail on the linear regression model in Fig. 2d/ Fig. S5 in the captions or text.

- Fig. 2e caption: The caption mentions the “Schiff base intermediate”, while in reality it is the conjugate acid of the Schiff base (the iminium ion, this is also shown in 2e).
- Fig. 3c: It is not entirely clear which distance is depicted here. Is the displayed distance in reference to the crystal structure as in Fig. S9 or was it calculated as in Fig. S10? Please add a brief explanation to the caption.
- Page 6, line 19: Michaelis-Menten parameters were determined for 30 (not 31 as written) of the retro aldolases according to Table S3 (5 were expressed but not determined, 1 did not express).
- Fig. S10: The colour-coding of the frames is not compatible with colour-blind inclusive design. I suggest replacement with or addition of an additional visual aid (e.g. symbol in the corner of the frames).
- Fig. S10: Please write out the abbreviated “NZ position” once in the caption.
- Methods: Please include the name of the plasmid which was used for cloning and in case of a not widely used plasmid, please include an entry number from a database/repository (e.g. Addgene). If a custom plasmid was used, it is recommended to submit the annotated sequence as an additional file or submitting it to a common plasmid repository and citing the entry. If it is not self-evident from the plasmid sequence, please also state which Golden Gate restriction sites were used.
- Methods: When reporting centrifugation steps (e.g. p. 11, line 1), the acceleration (in rcf or g) should be reported, not the rpm number because the latter is instrument-dependent. Duration and temperature of the centrifugation step should also be included. In the subsequent sentence about washing the pellet some information is missing, e.g. at which temperature were the pellets washed, how many times with which volume?
- Methods, p. 11, lines 23ff: Were the same protein concentrations used for the chemical denaturation experiments as in the previous paragraph about the thermal denaturation?
- Methods: The Python SciPy library was used (e.g. p. 11, line 28/p. 12, line 32). Please include a version number.
- Page 13, line 23: The number after 6-methoxy-2-naphthaldehyde “1” is probably for numbering the compound (but no other compound is numbered). If a reaction scheme is added (see minor comment above), additional compound numbers would be useful and should be formatted differently (e.g. in bold font).
- Page 13, line 24: Was the phosphate buffer’s pH adjusted to a specific value?
- Page 7, line 1ff: Here, the enantiomeric preference of only one design was investigated. What are the enantiomeric preferences for other constructs, e.g. the other highly active RAD35 or the active designs from which crystal structures could be obtained? Furthermore, the retro-aldol reaction was performed only with a racemic substrate – were different cleavage rates observed for the individual enantiomers (i.e. kinetic resolution of the racemic mixture)?
- Page 9, line 23ff: Please include a reference for CASP.
- Supplementary information: Please provide the 36 sequences chosen for the study in form of a table/ additional excel file or other adequate file format. Please include also sequences and used annealing temperatures of primers from PCR experiments. The annealing time for the PCR reaction is missing in the methods section.

(Remarks on code availability)

I am no expert in coding, so I have not reviewed the code. I assume you have other reviewers who cover that aspect.

Version 1:

Reviewer comments:

Referee #1

(Remarks to the Author)

The authors have clearly invested substantial effort in addressing the reviewer comments. They have added significant additional information and analyses (e.g., MD equilibration) which have considerably enhanced both the quality and comprehensiveness of the work. Additionally, the authors have improved the structure and documentation of the associated GitHub repository, in this way supporting reproducibility and usability of their in-silico workflows. I recommend the article for acceptance, subject to the following concern:

I strongly encourage the authors to remove the “original” and insufficiently documented RiffDiff repository (https://github.com/mabr3112/riff_diff_original), as it may cause confusion for users. Alternatively, they should clearly link to the better-documented repository (https://github.com/mabr3112/riff_diff_protflow).

Moreover, it is potentially problematic that the code is only made available under the umbrella of a new, previously

undescribed framework (“ProtFlow”), which is neither introduced nor discussed in the current manuscript. The RiffDiff code should be presented as a standalone, clearly defined methodological workflow.

In my view, the current mixture of acronyms, GitHub repositories, and workflow frameworks could hinder broader adoption of RiffDiff. Clarifying this aspect would substantially improve the overall accessibility and impact of the work.

(Remarks on code availability)

Please refer to general review comments.

Referee #2

(Remarks to the Author)

The authors have satisfactorily addressed most of my concerns; however, a few minor issues remain that should be addressed to improve readability and accuracy. The addition of MBHase designs has strengthened the paper, but some minor revisions in the description and interpretation of those results are needed for greater clarity. I recommend publication after all of the following minor comments are addressed:

Comment 1.

Abstract:

The authors claim that their “findings enable the practical applicability of de novo protein catalysts in synthesis and shed light on fundamental principles of protein design and enzyme catalysis.” However, the practical applicability of their enzymes for synthesis is not convincingly demonstrated. The retro-aldolases are not synthetically useful, and while the MBH reaction is valuable, the reported conversions ($\leq 16\%$) are too low to be considered practically useful. I recommend omitting this claim and rewording the abstract to instead highlight the key achievements: the design of de novo enzymes for two distinct reactions with higher activity than previous de novo enzymes, achieved by constructing a custom scaffold for the catalytic array rather than repurposing a natural scaffold.

Comment 2.

Page 2, line 32:

The statement “Since the first successful attempts, critical analysis of designed enzymes and their evolved variants revealed additional aspects of the enzyme design problem. One was the construction of catalytically potent theozymes” is unclear, as the term “catalytically potent theozyme” is not defined. A clearer phrasing might be:

“Since the first successful attempts, critical analysis of designed enzymes and their evolved variants revealed key challenges in enzyme design. One such challenge was the difficulty in predicting optimal theozymes for the reaction of interest.”

Moreover, the term “catalytically potent theozyme” remains unclear. What does it mean for a theozyme to be “potent”? Rather than referring to potency, it would be more accurate to state that the original theozymes used for the retro-aldol and MBH reactions were suboptimal, as evidenced by the fact that directed evolution replaced the designed catalytic arrays with alternative residues that enhanced catalytic efficiency.

For reference, I include below my original comment on this point:

“The term “catalytically potent theozyme” is also unclear. Is this potency based on theozyme DFT energy? Likely, the authors mean that the catalytic tetrad motif was empirically determined to be more efficient for the retro-aldol reaction than simpler motifs, such as the Lys/Glu pair used by Althoff et al. (Prot. Sci., 2012) to design RA95. Additionally, the tetrad with bound inhibitor used here is not a theozyme, as it is not a QM model of catalytic groups stabilizing a transition state. I recommend that the authors verify whether they are properly using the term theozyme throughout their article (see Tantillo et al., Curr. Opin. Chem. Biol., 1998).»

Comment 3.

Page 2, Line 23:

Please cite Privett et al. (<https://pubmed.ncbi.nlm.nih.gov/22357762/>), in addition to Siegel, Rothlisberger, and Jiang. It's important to acknowledge other pioneers in the field beyond just David Baker, including Steve Mayo.

Comment 4.

Page 3, Line 2:

The authors again describe “preorganization” incorrectly when they mean flexibility: “molecular dynamics simulations of retro-aldolases at intermediate steps during directed evolution demonstrated the importance of preorganization in catalytic arrays.” Preorganization and flexibility are not the same thing. This sentence should be reworded to avoid confusion.

For context, I paste below my original comment related to this topic:

“Preorganization Misinterpretation. The authors use MD to evaluate catalytic residue preorganization but MD, as applied here, cannot measure this. Preorganization refers to active-site groups not having to undergo significant rotation to stabilize the transition state, since they are already pointing in the correct direction (see Jindal & Warshel, Proteins, 2017). This can

only be shown by comparing active-site structures with and without a transition state (or analogue). Here, unbound structures were analyzed, so preorganization cannot be determined, as conformational changes may occur upon transition-state formation. The authors instead have assessed rigidity of the catalytic tetrad, indicating its ability to adopt productive or unproductive conformations, given the assumption that the designed configuration is catalytically productive. The authors should thus remove all discussion of preorganization from the manuscript and focus on rigidity/flexibility and/or productive/unproductive conformations. Along the same lines, on Fig. S9, it is stated that the catalytic tetrad is well preorganized. Notwithstanding the inability of MD as performed here to assess preorganization, the Y180 residue is shown to be highly flexible, which presumably would not be ideal for efficient catalysis.»

Comment 5.

Page 7, Line 4:

To facilitate analysis by reader, please indicate the fold-decrease in k_{cat} caused by mutation of tetrad residues (e.g., "Substitutions reduced k_{cat} by X–Y fold").

Comment 6.

Page 7, Line 34 and Fig. 3c:

Specify the linear regression model used and provide the equation. This is necessary for reproducibility.

Comment 7.

Page 8, Line 25:

Replace "can" with "could."

Comment 8.

Page 8, Line 31:

The unproductive conformation of Y120 is unlikely to result from crystal packing, as this residue is located within the active site rather than in a surface-exposed region where crystal contacts typically occur. Therefore, it is unlikely to be a crystallization artifact, as suggested in the text. More likely, this reflects an inaccuracy in the design model.

Comment 9.

Page 9, Line 22:

Please explain what is meant by "active-site metric". This term is unclear and should be defined.

Comment 10.

Page 9, Line 36:

The phrase "Streptavidin cofactor" is incorrect. Streptavidin is a protein, not a cofactor. Please clarify or revise this terminology.

Comment 11.

Page 10, Line 1 and Abstract:

The terms "active site constellations" and "active site arrays" are used interchangeably. Please standardize to one term throughout. I recommend "array", as "constellation" is not as common in enzymology literature.

Comment 12.

Page 10, First Paragraph:

Please add citations to Crawshaw (Ref 51) and Hutton (Ref 52) at the appropriate points in the discussion where their work is referenced.

Comment 13.

Suppl. Fi. S10:

Label each transition state (TS) and intermediate. This will help match the text (first paragraph on Page 10). Also, please explicitly show the MBHase mechanism involving Glu26 as acid/base in a separate panel to help readers clearly distinguish the two MBHase mechanisms discussed here.

Comment 14.

Page 10, Line 17:

Please indicate which transition state was used in the RiffDiff design of MBHases (refer to Suppl. Fig. S10).

Comment 15.

Page 10, Line 23:

Please provide geometric definitions (distances, angles, dihedrals) for the interactions between catalytic residues and TS (e.g., H-bonds, π - π stacking interactions, etc.) as a Supplementary Table or Figure. Also, please provide the cartesian coordinates for the theozymes used (e.g., in mol2 or PDB format) as Supplementary Files. This is required to enable others to reproduce the work.

Comment 16.

Page 11, First Paragraph:

MBH48 should be compared directly to BH1.8 23H (which shares the same catalytic array), in addition to BH32.8 (which uses a different array). For example:
"With a k_{cat} of 0.025 min^{-1} , MBH48 is 1.5 times more active than BH32.8 ($k_{cat} = 0.0168 \text{ min}^{-1}$), a variant that emerged after screening 13,590 clones over 8 rounds of directed evolution (Figure 5e). However, it remains 45-fold less active than BH1.8 23H ($k_{cat} = 1.13 \text{ min}^{-1}$), which shares the same catalytic array." This comparison is critical given the paper's emphasis on "catalytically potent theozymes" and underscores the need for future designs to incorporate features beyond precise catalytic array positioning. It also provides a more balanced perspective on the limitations of the current approach.

Comment 17.

Discussion and Outlook

This section contains several statements without appropriate citations. The discussion should situate the findings more clearly within the existing literature. Please add citations for the key statements listed below:

Page 11, Line 35: "Current enzyme design and engineering rely on high-throughput screening methods to produce viable enzymes."

Page 12, Line 23: "Our findings support the notion that designing enzymes with activities similar to their natural counterparts will require accounting for catalytic interactions and their conformational dynamics throughout the complete reaction cycle."

Page 12, Line 27: "Protein language models represent a new frontier for this task, but their poor generalizing abilities currently limit their effectiveness in novel chemical reactions, especially with de novo enzymes."

Page 12, Line 29: "Physics-based methods like QM/MM appear more suited to handle generalizability."

Page 12, Line 31: "An alternative route for physics-based activity prediction is conformational ensembles of the active site."

Page 12, Line 36: "This highlights how computational tools that optimize an active site's conformational ensemble could yield future improvements in activity of designed enzymes."

Page 13, Line 7: "Yet very few of these studies started from scratch and all those that did relied on directed evolution or screened several hundred de novo sequences to reach practically useful activities in the designed biocatalysts."

Comment 18.

Please make Suppl. Fig. S5 bigger, including fonts. It is hard to read.

Comment 19.

Fig. 3f: ee values in bar graph don't match the legend.

Comment 20.

Fig. 4: Structures are very similar but not almost identical (1.2 Angstrom RMSD is not almost identical). Please reword. Also, the figure legend describes a theozyme for panel c but no theozyme is shown, only catalytic array.

With these revisions, the manuscript will present its findings more clearly and accurately and will better position itself within the context of existing work in enzyme design and catalysis.

(Remarks on code availability)

It works. We are using some aspects of it in our research.

Referee #3

(Remarks to the Author)

The authors have addressed nearly all of my previous comments to full satisfaction and even exceeded expectations in answering remaining questions. Especially with the additional challenging application on the abiological Morita-Baylis-Hillman reaction, the general applicability and thus potential impact of the new de novo enzyme prediction pipeline is now well supported. I therefore fully support publication of the work.

I recommended the following minor adjustment for the final manuscript:

1. One of my comments on the initial draft concerned the type of replicates (comment point 5), mainly to include biological duplicates for major constructs. The authors state in the reworked manuscript that biological duplicates were performed in the methods section when describing protein expression. In the rebuttal document the authors also say that the data from biological replicates is part of the analysis already in the reworked manuscript. The description of data (e.g. Michaelis-Menten analyses) do not clearly reflect this. In figure captions it is still only stated that the data stem from "triplicates", implying the figures represent single biological replicates of technical triplicates. The data of biological replicates can be included as supplemental material but should then be cross-referenced where needed. If the biological duplicates of the triplicate measurements are already part of the data, the figure descriptions should reflect this to avoid confusion.

2. Several figures (1c, 3d, 4c,d, 5e, S3 and S14a) are still very hard to interpret for people with colour vision impairment,

which becomes apparent when viewed as a grayscale document. For the respective figures I suggest adapting to a colour-blind palette or in case of line charts, adding different texture to the lines (e.g. dotted lines, etc.).

3. Methods section in main document, p. 22, l. 4 typo: "p 7.4" instead of "pH 7.4"

(Remarks on code availability)

Referee #4

(Remarks to the Author)

I co-reviewed this manuscript with one of the reviewers who provided the listed reports.

(Remarks on code availability)

Version 2:

Reviewer comments:

Referee #1

(Remarks to the Author)

I appreciate that the authors removed the old and poorly documented repository and agree that the "protflow implementation" of Riff-Diff is now well-documented enabling everybody from the target audience to execute the pipeline effectively.

(Remarks on code availability)

Referee #2

(Remarks to the Author)

The authors have addressed all my comments satisfactorily.

However, I noticed that the mass spectrometry methods are not included in the Methods section. Was the analysis performed by MALDI? Which instrument was used? Please provide the missing methodological details.

Congratulations on your nice work!

(Remarks on code availability)

Referee #1:

In the research paper "Computational Design of Highly Active De Novo Enzymes," Oberdorfer et al. introduce Riff-Diff, a method for de novo enzyme design that transplants catalytic arrays into newly generated protein backbones. The method incorporates artificial motif libraries by embedding catalytic arrays into alpha-helical fragments, which are then scaffolded using RFDiffusion. To improve binding pocket formation, a custom auxiliary potential and placeholder alpha-helices were used, leading to optimized pockets without relying on substrate auxiliary potentials.

Riff-Diff's iterative refinement cycles further optimize backbone-side chain interactions, guided by Rosetta's FastDesign protocol and ProteinMPNN for sequence design. The method was applied to retro-aldolases, successfully scaffolding a catalytic tetrad, resulting in designs with catalytic rates exceeding five-millionfold acceleration over the uncatalyzed reaction.

Of the 35 successfully cloned designs, all led to soluble proteins, with many showing measurable catalytic activity. Structural and biochemical validation, including SAXS, CD spectroscopy, and mass spectrometry, confirmed structural integrity and thermal stability of the designs.

Overall, this study presents an application of established computational design tools to scaffold catalytic arrays within de novo protein backbones. The authors employed key existing methodologies such as RFDiffusion, ProteinMPNN, and Rosetta FastDesign, combining them effectively to create a small validation library of retro-aldol enzymes. The research also includes important biochemical validation steps to assess the activity and structural stability of the designed enzymes.

The work demonstrates skillful use of cutting-edge protein design tools in a context that shows potential, though it is important to note that the core methods (i.e., RFDiffusion, ProteinMPNN, and Rosetta FastDesign) are not novel. Furthermore, the claim of generating "highly active enzymes" is not entirely supported, as there is no transparent comparison with state-of-the-art de novo enzymes or artificially evolved variants, making it challenging to fully evaluate the method's impact. Additionally, the approach is limited to cases where known catalytic motifs are available, restricting its generalizability for novel enzymatic functions.

We sincerely thank referee #1 for their kind words about the manuscript and want to emphasize our appreciation for the thorough summary provided. We especially thank them for the comment about transparency, which prompted us to include two separate tables in the supplementary materials, listing all previous design and evolution efforts for retro-aldolases.

In addition, we want to emphasize that RiffDiff does not require prespecified rotameric states for active site residues. These are generated during artificial motif generation. In essence RiffDiff splits the problem of identification of non-clashing primary transition state interactions and backbone generation into two separate steps. This is ideal, since the focus during backbone

generation can be put on binding pocket/entrance channel and second shell interactions – a necessity for successful enzyme design campaigns. We believe that this point was not conveyed well in our initial submission and thus added a detailed description and explanation of RiffDiff to the supplementary materials.

We made the code for RiffDiff in its original form and as part of a python package, which we are developing in-house for all our protein design efforts, available on github. The links are provided for reference below and also included at the end of the main text. The corresponding design model and data files have been uploaded to Zenodo.

RiffDiff (original): https://github.com/mabr3112/riff_diff_original

RiffDiff (implemented in ProtFlow): https://github.com/mabr3112/riff_diff_protflow

Zenodo DB: <https://zenodo.org/records/15494858>

Despite these limitations, this work holds relevance for researchers in enzyme design and biocatalysis but I feel it may be better suited for a specialized journal like Nature Catalysis after addressing the concerns outlined below.

Novelty of the study

The novelty of this study is limited by its heavy reliance on previously established tools and methods, such as RFdiffusion, ProteinMPNN, and Rosetta FastDesign. These technologies have already demonstrated success in key areas of protein design, including scaffolding catalytic sites, designing protein monomers, binders, and symmetric assemblies. The RFdiffusion model has been extensively validated for a broad range of design challenges, such as topology-constrained design, metal-binding protein creation, and enzyme active site scaffolding. As a result, while the study claims to achieve high activity in de novo enzymes, it primarily refines existing methods without introducing fundamentally new concepts or frameworks for protein design.

Thank you for pointing out that we've largely used previously established tools. However, we must politely disagree on the notion of the above paragraph. We don't think that it is necessary to have a new design algorithm to provide fundamental insights. In fact, we hope that our manuscript shows that combination and modification of established methods and careful analysis can lead to practically useful and generally applicable implementations. We are convinced that the introduction of our artificial motif libraries, the addition of the custom potential to RFdiffusion and the introduction of a place holder helix as well as the iterative ML and atomistic modelling design steps significantly improve the success rates of the described design efforts and in all described cases are responsible for the – comparatively – high activity. We also highlight some of the detailed investigations that led us to implement RiffDiff the way it is in Figure 1c of the main text. Here we show how 'vanilla' RFdiffusion is not reproducing 'natural' enzyme pockets and how this can be alleviated with running RiffDiff.

Comments on the manuscript

General:

1. A direct comparison with other de novo designed retro-aldolases is essential and should be included to establish a baseline for evaluating the success of this study. Additionally, the authors need to substantiate their claim that their "highly active de novo enzyme" rivals those optimized by in vitro evolution by providing a direct comparison of catalytic performance under the same experimental conditions (K_m , k_{cat} , k_{cat}/K_m , TTN etc). In this respect, the captions for Figs 1 and 2 contain strong claims that are not well supported by the presented data and illustrations. For instance, the caption for Fig. 2 states, "Activity of designed retro-aldol enzymes far exceeded those of previous zero-shot designs." However, the figure does not clearly convey this observation. I suggest that the authors include a table and a bar plot that explicitly presents the k_{cat} and K_m values for: a) the best-performing de novo designs from the literature, b) the best-performing engineered variants from the literature, and c) the RiffDiff variants.

This is an excellent suggestion, and we included a table detailing past designed and evolved retro aldolases and enzymes for the Morita-Baylis-Hillman reaction into the supplementary materials. Figure 3 in the main text details the activities in comparison to previous computationally designed retro aldolases and the variants along the evolutionary trajectory of RA95.5-8F. Figure 5 compares k_{cat} values for designed and evolved MBHases.

2. Do the authors anticipate that their method can be applied in cases where no functional catalytic arrays are known?

This is an excellent question, and we realized that we did not explicitly discuss this in the original draft. We think that, like all other currently available methods, RiffDiff works as long as there is an active site model available. Whether this comes from naturally occurring proteins, or from DFT/QM calculations does not matter. We now comment on this specifically in the second paragraph of the conclusions.

3. Page 2, line 11: I do not agree that certain reactions are completely unsuitable for high-throughput screening. While it is valid to acknowledge that time and resources can sometimes be limited, or that the product may not justify the screening effort, in principle, every reaction could be screened at a higher throughput. This ultimately depends on the effort invested in developing analytical methods. Further, the authors do not mention numerous computational strategies other than de novo design have been developed to reduce screening effort.

<https://www.sciencedirect.com/science/article/pii/S0959440X21000154>

<https://www.nature.com/articles/s41592-019-0496-6>

<https://www.nature.com/articles/s41589-024-01712-3>

Thank you for pointing this out. We agree that our initial statement was too narrow and revised the corresponding section in the main text following the reviewer's suggestion and rephrased "inaccessible" to "difficult to access".

4. Page 2, lines 35-37: Next to retro aldolases, decades of efforts on de novo design and optimization of Kemp Eliminase should be acknowledged in a sentence providing suitable citations. Especially because Kemp is later used as illustrative example for discriminating between active and inactive designs using MD simulation (line 45 and following).

<https://www.sciencedirect.com/science/article/pii/S0022283611000842>

<https://www.nature.com/articles/nature12623>

<https://www.nature.com/articles/s41467-020-18619-x>

We apologize for missing this and added the corresponding citations to the Kemp eliminase example.

5. Page 3, line 1: Next to the relevance of conformational preorganization on protein level, MD simulations studying enzyme-substrate interactions have been demonstrated to provide relevant insights in the context of optimized de novo enzymes.

<https://pubs.acs.org/doi/full/10.1021/acs.jcim.3c00002>

Thus, I recommend that the authors also study their designs using MD simulations of enzyme-substrate complexes rather than solely focusing on protein level preorganization.

We want to thank the reviewer for this comment and agree about the potential added insights of MD simulations of enzyme-substrate complexes. In the absence of reliable experimental structures of enzyme-ligand complexes, we used AlphaFold3 to model and evaluate enzyme-ligand interactions. The additional findings are integrated into the revised main text as a standalone paragraph next to the MD section. We are convinced that this data is enhancing our study further.

6. Page 4, lines 4-8: It is not fully clear why, where, and how the authors added the additional alpha-helix. How did they decide on length and geometric properties of this helix? In general, how did you select the sequence and structures of the helical fragments?

We apologize for not being clear enough here. In the revised version, a detailed description about how the alpha-helix is positioned was added to the methods section. Detailed description on picking helical fragments is also included in the methods section and supplementary materials.

7. Page 4, line 24: Please explain why you used the underperforming ESMFold instead of AlphaFold2 in this step – tradeoff between speed and accuracy? Later (line 29) you use AlphaFold2.

Thank you for asking this question. Indeed, we use ESMFold for its speed, and we added this clarification into the main text.

8. Page 5, lines 11-12: Please specify how you analyzed the buried active site and how you defined which binding site is buried and which not.

Apologies for not being clear enough about this. Just from the way RiffDiff operates, it basically ensures that the active site is 'buried'. What we meant here though is that we excluded designs based on accessibility of their active site – e.g. if channel occluding mutations were introduced. This was done based on manual inspection and this is now reflected in the text.

9. Page 5, lines 11-12: Please specify what you mean with “most” (e.g. 20/35) and how “measurable” was defined.

Thank you for pointing out unscientific language. We exchanged these phrases with concrete numbers.

10. Page 5, line 15: I understand that structural similarity (TM-score) is the most relevant metric. In addition, it would be interesting how similar the closest natural sequences are (sequence identity, sequence similarity, coverage).

Very valid point! We performed a blast search of the designed sequences against the non-redundant NCBI sequence database, which resulted in no significantly similar hits (E-value threshold = 0.05; default). This is now explicitly stated in the main text.

10. Page 6, lines 1-2: I encourage the authors to also determine protein melting temperatures (T_m) as a measure for stability.

We agree that these are valuable data point and show the thermal melting curves in Figure 3a of the revised manuscript. We describe the thermal stability of the expressed variants with CD thermal scans up to 95 °C.

11. Page 7, line 4: Please explain how you defined “partially functional”.

We agree that this is not defined at all and apologize for this inaccuracy. Since we restructured this whole section, due to the addition of additional data, we re-wrote the commented section entirely. This referenced statement is now reflected with ‘Seven designs exceeded this range, suggesting their tetrad residues might participate in the catalytic mechanism’

12. Page 7, line 24: It is very common that there is a high deviation between MD simulation replicates, for which reason several replicates should be performed (at least 5, better 10-20). In the context of Kemp eliminase it has been shown that many short simulations can help to achieve the necessary throughput if computational resources are limited. Please see: <https://pubs.acs.org/doi/full/10.1021/acs.jcim.3c00002>

We appreciate the reviewer’s comment and agree that more MD simulation replicates can be crucial. Thus, we performed MD simulations of all designed enzymes and of RA95.5-8F in 20 replicates. The corresponding detailed analysis has been added to the manuscript in a new

section detailing active site dynamics and uncoupling of precision and positioning of active site residues. We want to thank the reviewer for this comment again, as we believe the additional analysis significantly improved the findings of our study.

13. Page 10, lines 9-10: Please specify what you used as positive and negative control.

Apologies for not clearly pointing this out. This is now specified in the text.

14. Page 10, line 13: I suggest performing triplicates instead of duplicates.

Great point! The commented section describes the expression procedure for the initial activity screen. Because the purpose of this procedure was to quickly screen variants for activity, we did not see the necessity to perform the screen in triplicates, as this would have complicated the experimental setup (more 96-well plates and not enough ‘deck’ positions on our liquid handling robot). Since all active designs have been studied in depth afterwards, we think this is fine.

15. Page 10, line 41: Why did you switch from 50 to 100 mg/L kanamycin?

We sincerely appreciate the reviewers detailed comment! The switch has been made at some point during the last three years. This originated during a phase in which we wanted to have a more stringent selection of positive clones. We kept this protocol ever since in our lab.

16. Page 11, lines 3-4: DNase & lysozyme concentrations are missing.

This information has been added to the methods.

17. Page 11, line 9: Do you mean $1/16 = 0.062$ mg?

Yes. Sorry for the confusing notation. The exact amount is now specified.

18. Page 11, line 12: Please specify the ingredients of the storage buffer.

The contents of the storage buffer are now described in the Methods section. Here for reference: 20 mM NaPi/300 mM NaCl/2 mM TCEP pH 7.4 for RAD designs, 20 mM NaPi/150 mM NaCl pH 7.4 for MBH designs

19. Page 11, line 16: Circular Dichroism (CD) and thermal denaturation

We apologize for not spelling this out. We made sure that all abbreviations in the main text and supplementary materials are spelled out at least once without abbreviation.

20. Page 12, line 7: Please specify what you used as precipitant solution.

We added the exact contents of the precipitant solution to the crystallography table in the supplementary materials.

21. Page 12, line 30: Please specify the full range and step size of the serial dilution.

The missing information was as added to the Methods section.

21. Page 13, line 2: In my opinion an introductory sentence specifying Gromacs version and purpose of the experiment is necessary.

We agree that this is helpful and added the GROMACS version, and a description of the purpose of the MD experiment to the MD section of the revised main text.

22. Page 13, lines 12-13: NVT and NPT equilibration seem very short for de novo designed structures. Please provide quality measures of the equilibration success (RMSD, radius of gyration, potential energy, temperature, and pressure). It's common to apply positional restraints to the heavy atoms of the backbone or specific regions of the protein during the initial stages of equilibration to prevent large conformational changes while the solvent is equilibrating. If restraints were used, specifying the force constant and the atoms to which they were applied would be beneficial.

Good suggestion! We have now added additional information on the equilibration of the structures for MD to the revised main text methods and to the supplementary materials (Supplementary Figure S16)

23. Page 13, line 18: Using the LINCS algorithm for bond constraints is standard practice, particularly for systems involving hydrogen atoms. However, clarifying which bonds are constrained (e.g., all bonds involving hydrogen) is necessary.

Thank you for making us aware of this. We now specified that hydrogen bonds are constrained.

24. Please doublecheck the clashes, the number of ligand and solvent molecules and ensure that all units (e.g. for angles and distances) are specified in the SI.

We have double-checked the clashes and the number of ligand and solvent atoms. All clashes in the PDB files are minor and justified by the experimental data (i.e., electron density map). The values reported as "ligands" and "solvent" in Table S10 are correct and represent the number of atoms, not molecules. We thank the Reviewer for pointing this out

25. The license under which the code can be used by others should be specified (e.g. MIT).

Our apologies for not explicitly mentioning this. We added an MIT license to the repository.

26. Documentation on a) how to install dependencies, b) needed hardware and software requirements, c) installation and primary citation of external tools, and d) workflow and how to use the provided code is completely missing.

We are incredibly sorry about this shortcoming and apologize for not providing clear guidelines on who to use RiffDiff. We are convinced that tools like this will have much higher impact if they are easy to access and use. Therefore, we implemented RiffDiff in the python package ProtFlow (https://github.com/mabr3112/riff_diff_protflow), which can run the scripts on SLURM-based HPC clusters and on local machines. The repository now contains detailed instructions for use, together with example input and output.

27. I strongly recommend providing some kind of workflow script or jupyter notebook; the repository contains more than 10 separate Python scripts without any documentation on how and in which order to use them.

As mentioned above, we added a comprehensive installation and workflow guide into the tools repository.

28. I strongly recommend providing a Docker container or at least conda yml files with the needed software.

We agree 100%. The above-mentioned python package, ProtFlow, contains conda yml environment files that are required to run Riff-Diff.

Given that the primary contribution of the authors' work is a novel method for the de novo design of efficient enzymes, it would be highly beneficial to provide a comprehensive, reproducible workflow or tool to facilitate broader use. This is essential for the technology's adoption and for maximizing its impact on future research. While I recognize the expertise shown in utilizing external software and tools, the innovative value of RiffDiff as a new technology or methodology would be significantly enhanced if it were made usable for other researchers.

Again, we want to thank the reviewer for this comment and agree to 100%. Please see the answers and explanations above.

Referee #2:

In this article, Oberdorfer and colleagues introduce RiffDiff, a de novo enzyme design pipeline that constructs enzymes for specific reactions by building an artificial protein scaffold around a designated catalytic motif with its associated ligand. RiffDiff begins with the side chain coordinates of a preexisting catalytic motif extracted from the PDB—here, the catalytic tetrad of evolved retro-aldolase RA95.5-8F—embedding each tetrad residue on its own alpha-helical peptide to determine compatible positions of backbone atoms that avoid steric clashes while maintaining the catalytically productive arrangement of functional groups. The method then fills backbone gaps between these peptides using RFDiffusion, designs sequences to stabilize the scaffold using ProteinMPNN, and refines active-site rotamer configurations using the FastDesign and CoupledMoves protocols implemented in the Rosetta protein design suite.

Using RiffDiff, the authors designed 36 retro-aldolases, 30 of which displayed activity. The most active designs are the most efficient de novo retro-aldolases reported to date, though their catalytic rates remain modest ($k_{cat} \leq 0.036 \text{ s}^{-1}$). Crystal structures for four designs confirm the accuracy of the designed active-site configurations and/or overall folds. The authors conclude the article with a discussion offering insights and suggestions for further refinement. This study demonstrates impressive results, and several aspects of RiffDiff's methodology, such as its use of an artificial motif library and placeholder alpha-helices for entry-channel design, are particularly innovative. RiffDiff promises to become a valuable tool in de novo enzyme design, complementing the traditional theozyme-based approach. I recommend publication of this article following revisions to clarify methods, temper unsupported claims, and add important missing data.

We sincerely thank referee #2 for their kind words about our study and method and want to emphasize our appreciation for the thorough summary provided. We especially thank them for the comment about the catalytic rates, which prompted us to change the title to 'Computational enzyme design by catalytic motif scaffolding'.

1. Overstatements and Misleading Claims. The title of this article is misleading. It implies high activity even though these enzymes are not highly active despite being the most active computationally designed retro-aldolases to date. For example, the most active design, RA29, has a catalytic efficiency of $290 \text{ M}^{-1} \text{ s}^{-1}$, which is orders of magnitude lower than the average natural enzyme (k_{cat}/K_M of $\sim 100,000 \text{ M}^{-1} \text{ s}^{-1}$) or even the most active artificial retro-aldolase (RA95.5-8F, $k_{cat}/K_M = 34,000 \text{ M}^{-1} \text{ s}^{-1}$). This value is instead comparable to those of other de novo enzymes, such as Kemp eliminases HG3 ($k_{cat}/K_M = 1300 \text{ M}^{-1} \text{ s}^{-1}$, see Blomberg et al., Nature 2013) and KE59 ($k_{cat}/K_M = 160 \text{ M}^{-1} \text{ s}^{-1}$; see Khersonsky et al., PNAS, 2012). Furthermore, k_{cat} values are modest, with a maximum of 0.036 s^{-1} . These results indicate that RiffDiff produces de novo enzymes with modest activity, contradicting the title's claim that the enzymes are "highly active". Thus, the title should be revised. The key achievement of this paper is not that the enzymes' activity is high, but that de novo enzymes can be constructed by building a completely new protein to scaffold a catalytic motif and form a binding pocket for

the ligand—without relying on existing protein scaffolds. This point, rather than overstating activity, should be reflected in the title. The abstract claims that the “designs exhibit a high fold diversity”. Yet, all designs are single-domain, alpha-helical proteins, which doesn’t suggest high diversity. The authors likely mean that the designs adopt distinct folds; this should be clarified to avoid confusion.

We thank the reviewer very much for the comment and the important points raised! We agree that the statement in the title was exaggerating and like to emphasize that this was meant in relation to other computationally designed de novo enzymes and in hindsight is very different for various reactions and thus probably always underspecified. Thus, we changed the title to ‘Computational enzyme design by catalytic motif scaffolding’, as already mentioned above. The revised version now also talks about designed enzymes exhibiting ‘distinct’ folds rather than diverse.

The claim of “proficient retro-aldol enzymes” in the abstract is misleading. Although these enzymes are the most active computationally designed retro-aldolases to date, they are not proficient biocatalysts (as discussed above).

We agree with the reviewer’s sentiment and refrain from using this claim in the abstract.

2. Preorganization Misinterpretation. The authors use MD to evaluate catalytic residue preorganization but MD, as applied here, cannot measure this. Preorganization refers to active-site groups not having to undergo significant rotation to stabilize the transition state, since they are already pointing in the correct direction (see Jindal & Warshel, *Proteins*, 2017). This can only be shown by comparing active-site structures with and without a transition state (or analogue). Here, unbound structures were analyzed, so preorganization cannot be determined, as conformational changes may occur upon transition-state formation. The authors instead have assessed rigidity of the catalytic tetrad, indicating its ability to adopt productive or unproductive conformations, given the assumption that the designed configuration is catalytically productive. The authors should thus remove all discussion of preorganization from the manuscript and focus on rigidity/flexibility and/or productive/unproductive conformations. Along the same lines, on Fig. S9, it is stated that the catalytic tetrad is well preorganized. Notwithstanding the inability of MD as performed here to assess preorganization, the Y180 residue is shown to be highly flexible, which presumably would not be ideal for efficient catalysis.

We highly appreciate this comment and apologize for confusing the terms. As suggested, we changed the evaluation and discussion of MD results on rigidity/flexibility and productive/unproductive conformations. To substantiate our analysis further, we performed additional simulations and increased the number of MD replicates from 3 to 20. The flexibility of Y180 in the original simulations of RA95.5-8F was caused by using an initial model as input for the MD simulations that had Y180 in a conformation that was flipped outward of the active site. This was fixed in the new simulations.

3. Conclusions not fully supported by data. The statement “High activity originated from catalytic tetrad” (section title, p. 6) is not fully supported by the data. While the authors mutated

Lys to Ala to demonstrate the importance of the designed nucleophile, they did not mutate the other tetrad residues to confirm their individual contributions to catalysis. Without this data, it is unclear whether designing the entire tetrad was necessary or if a subset of these residues would suffice. To substantiate their claims regarding the catalytic tetrad's role, the authors must create knockouts for each tetrad residue in a subset of their enzymes and report the resulting k_{cat} and K_m values (as done by Obexer et al., Nat. Chem., 2017). On p. 7, the authors state that the preference of "RAD29 to form (R)-methodol in the forward reaction (aldol addition) further supports the successful design of a specific substrate binding mode and participation of all tetrad residues". This is incorrect: the reported 60% enantiomeric excess indicates an 80:20 R:S ratio, demonstrating that the active site is not specific to a single substrate binding mode. This is especially evident when compared to RA95.5-8F, which achieved an enantiomeric excess of 99.2:0.8 (Obexer et al., Nat. Chem., 2017). Furthermore, this result does not prove the participation of all tetrad residues in catalysis.

The reviewer raises excellent points, and we agree that our initial statements need further, corroborating experiments. As suggested, we performed site-directed mutagenesis experiments and integrated the corresponding results and their thorough analysis into the section discussing the catalytic activities of the designed enzymes, in the revised version of our manuscript. Moreover, we performed additional experiments on biotransformations in the aldol- and retro-aldol direction and rephrased the statement about the enantio-selectivity of RAD29 in the revised version.

4. Methods Section Completeness. The methods section lacks essential details on the computational design procedure, which is necessary for reproducibility. Missing specifics include: probabilities of catalytic tetrad rotamers used to generate the artificial motif library (and associated ϕ/ψ bonds), details for energy calculations, residue identities allowed at active-site positions during FastDesign/CoupledMoves (i.e. the searched sequence space), definitions of ligand interactions (e.g., geometric definition of contacts), amino acid probability cutoffs used to select sequence space for active-site optimization, inhibitor attributes (including treatment of covalent bond with Lys), design filtering criteria (e.g., predicted side-chain RMSDs), etc. Authors must include a detailed description of RiffDiff in the methods section and provide scripts or code for using it. The article should not be published without these. Page 4, line 29 mentions that final sequences were evaluated with AlphaFold2 and ranked using metrics; these metrics should be specified.

We highly appreciate this comment and agree that important details were absent from the method section. In the revised version, we answer all the raised questions of the above paragraph and included a detailed and step-by-step explanation of RiffDiff to the beginning of the supplementary materials.

Please clarify the term "free folding energies"; does this refer to a computed potential energy difference between the folded structure and an unfolded one? Or is this a true free energy that includes entropy? How is this calculated?

This is a great question, and we apologize for being unclear here. This term merely refers to the total score as calculated by the Rosetta Energy Function. To avoid any confusion, we clarified this in the revised version of the manuscript.

In general, the methods section as a whole should be revised to enhance clarity and provide sufficient details for reproducibility, including descriptions of negative/positive controls, plasmids, promoters, enzyme units used for kinetic assays, extinction coefficients for concentration calculations, chemical purity and source, etc.

This is a great suggestion, which was also raised by the other reviewers. We revised the methods section following the comments of all reviewers.

5. Data Presentation. The most important data in this article are the kinetic parameters (k_{cat} , KM) and thermodynamic stabilities of the designed enzymes, which are currently buried in supplementary materials. Please consolidate these data into a single table in the main text and convert C_m values to ΔG to provide more meaningful insights.

We highly appreciate the reviewers view of our results and are happy to say that we think alike and that the provided wealth of activity data will be a treasure trove for future studies. However, we believe a table containing k_{cat}/KM /denaturation midpoints for 36 constructs cannot be properly integrated into the main text, as it would take a full page on its own. We included a table containing these values in the supplementary materials. Moreover, we want to highlight that Figure 2e of the main text contains a comparison of k_{cat} 's in our designs with previously reported designed and evolved retro-aldolases.

6. Missing pKa Values. Please include pKa values for the catalytic lysine, at least for the most active designs (RA29/RA35) and those with available crystal structures (RAD13/RAD17/RAD32/RAD36). The discussion of catalytic tetrad activity is incomplete without these, since low k_{cat} could in part be due to high pKa.

This is an excellent suggestion, and we thank the reviewer for pointing this out. We performed pH-rate dependence assays and discuss them in the main text and show a table in the supplementary materials. We believe this significantly supports our findings and strengthens the study.

7. Clarifications needed. What do the authors mean by “zero shot”? From what I understand, zero shot is used to describe a machine learning model's ability to perform a task it hasn't been explicitly trained on. Yet, the ML methods used in RiffDiff (i.e. RFDiffusion and ProteinMPNN) were specifically trained to design protein structures and sequences, which is what they're used to do here. It is important to note that in this article, catalysis is not explicitly designed, since there is no QM calculation used to predict activity. Instead, catalysis is inferred from the sequence and structure of the active site, making it unclear why the term zero-shot is used here.

This is a valid point! We changed the term to 'one-shot' to reflect that the enzymes were designed computationally and the activities were achieved without a design-build-test-learn feedback cycle.

The term "catalytically potent theozyme" is also unclear. Is this potency based on theozyme DFT energy? Likely, the authors mean that the catalytic tetrad motif was empirically determined to be more efficient for the retro-aldol reaction than simpler motifs, such as the Lys/Glu pair used by Althoff et al. (Prot. Sci., 2012) to design RA95. Additionally, the tetrad with bound inhibitor used here is not a theozyme, as it is not a QM model of catalytic groups stabilizing a transition state. I recommend that the authors verify whether they are properly using the term theozyme throughout their article (see Tantillo et al., Curr. Opin. Chem. Biol., 1998).

Thank you for making us aware of this potentially misleading statement. In fact, for the retro aldolases, the scaffolded tetrad was indeed not a theozyme as defined in the original literature, but a catalytic array taken from an experimentally determined crystal structure of an laboratory evolved enzyme. We refer to the tetrad now as "tetrad" throughout the manuscript.

The authors analyse crystal structures to explain activity differences between variants. They focus on analysing the accuracy of the catalytic tetrad coordinates between design model and crystal structure. Yet, they do not discuss whether the binding pocket itself is amenable to efficient binding of the substrate. From looking at the crystal structures, I can see that not all of them have a well-defined binding pocket for the substrate. In the absence of crystal structures with a bound substrate analogue, the authors should perform docking to verify whether the substrate binding pocket can make interactions with this ligand as designed. It may also help to explain the reported ee.

We thank the reviewer for this detailed observation. To further corroborate our results, we performed AF3 predictions of the enzymes in complex with (R)-Methodol and in the hemiaminal bound intermediate state. The predictions are described in a completely new section of the revised main text with the subheading 'Active site dynamics reveal uncoupling of precision and positioning'. In this new section, we integrated findings from MD simulations and the AF3 predictions.

The enantiomeric excess of 60% obtained for synthesis of methodol by RAD29 indicates an 80:20 preference for the R-enantiomer, which suggests that the aldehyde substrate binds in multiple orientations within the active site. This result challenges the notion that a binding pocket complementary to the substrate has been designed. Given that the design procedure used a placeholder helix during RFDiffusion, is it possible that removal of this helix created a pocket that was not ideal for the substrate? The authors should discuss this potential limitation of the method.

This is a great suggestion and a valid point! We agree that removing the helix can lead to the design of an unspecific binding pocket, which is a source of low selectivity and potentially a

concern. However, the helix is removed directly after the initial backbone generation step and the pocket and interactions with the substrate present are designed subsequently during the following backbone refinement protocol and the CoupledMoves steps. Thus, we do anticipate high shape complementarity between substrate and binding pocket. To further clarify this point, we added a detailed description of the full RiffDiff protocol to the beginning of the supplementary materials.

8. Missing References. Page 2, line 20: “minimal active sites, called theozymes, can be constructed by placing amino acid functional groups around a computational model of a transition state in a stabilizing geometry.” The reference cited here (Zanghellini et al., *Prot. Sci.*, 2006) is not appropriate, as it describes the RosettaMatch protocol instead of the concept of a theozyme. You should instead cite the work of Tantillo & Houk on defining theozymes (*Curr. Opin. Chem. Biol.*, 1998).

We agree that this is the wrong reference and apologize for this mistake. It has been corrected.

Page 3, line 4: “Despite all these findings, recent de novo enzyme design strategies rarely optimized preorganization explicitly.” There is a recent example (Rakotoharisoa et al., *JACS*, 2024) where preorganization was optimized during the enzyme design procedure and experimentally validated using crystal structures with and without a transition-state analogue. This work should be cited.

We thank the reviewer for pointing this out and reworked this section in the revised version.

9. Missing data. Please provide amino acid sequences of all 36 designs in a FASTA format. Please provide PDB files for all design models, with inhibitor bound (since these were designed with Rosetta).

We included the sequences of all ordered designs (RAD and MBH) in the supplementary materials and uploaded the corresponding PDB files, containing the models to Zenodo (<https://doi.org/10.5281/zenodo.15494858>).

Please include a figure that shows the electron density of catalytic residues, to confirm that these were properly modelled. Inspection of the RA36 structure suggests that the tetrad Asn and Lys residues could benefit from improved modeling.

As suggested by the Reviewer, we have included a figure (Figure S15) displaying the electron density of the catalytic residues for each crystal structure.

Regarding RAD36, we invested much time and effort to identify the most probable conformation of the catalytic tetrad, especially Lys69, using different approaches like enhanced

maps (i.e., feature-enhanced maps and maximum entropy maps) and ensemble refinement in Phenix. Feature-enhanced maps ultimately provided sufficient electron density to model three conformations for Lys69. Based on the 2mFo-DFc map (see Coot screenshot below), we understand why the impression arises that Asn178 and Lys69 were suboptimal modelled. For example (see screenshot below), the amino group of one Lys69 conformation does not exhibit any electron density despite the available difference electron density (green density) next to it. However, moving the amino group into this difference density causes a significant clash with Asn178. Therefore, we concluded that the difference electron density between Asn178 and Lys69, which is rather spherical, might correspond to a water molecule that is located there when Lys69 adopts a conformation distant to Asn178. We did not model this water molecule to facilitate refinement and avoid confusion. As the calculated feature-enhanced map supports our modelled Lys69 conformations, we have added a panel to the new Figure S15 that shows the feature-enhanced electron density for Lys69 in RAD36.

We also revised the associated text in the Methods section (see revised Supporting Information) to clarify that we have also used feature-enhanced maps for the RAD36 dataset.

10. Minor comments. p. 3, line 42: “high specificity” should be replaced by “narrow specificity”. High specificity refers to a high k_{cat}/K_M value, whereas narrow specificity is used to describe an enzyme’s ability to react with only a small number of substrates.

We highly appreciate clarifying the difference here! We removed the word high.

p. 4: "vanilla RFDiffusion" needs clarification—is this a specific version? If so, specify version or release date.

This is indeed undefined. We removed the term "vanilla" and are addressing default RFDiffusion as ‘RFDiffusion’, without anything else.

p. 5, line 10: Name the ligand involved in binding-pocket interactions.

This is a great suggestion. In fact, we now mention the ligands/substrate by number, after they have been introduced for the first time, throughout the revised manuscript.

Avoid using "exceptional" to describe model accuracy, as this is subjective. Values of 0.89 Å and 1.2 Å RMSD, while accurate, are not "exceptional."

We appreciate making us aware of subjective wording and thus removed terms like exceptional in the revised version.

Typos: replace "lysins" with "lysines"; p. 9, line 1: "catalytic site" should be "catalytic motif."
We doublechecked the manuscript for typos and hope we fixed all spelling mistakes.

Fig. S6: Confirm if this shows racemic methodol.

We clarified in the text and all figure captions that rac-methodol was the substrate.

Indicate error and number of technical/biological replicates for all figures.

This is clearly stated and visible now. We included error and number of replicates in the methods and figure captions.

Table S4's title reads like a footnote, please fix.

All figure captions and table titles have been updated and should adhere to the same styling format.

Fig. 1c, middle graph: Improve line visibility.

Fig. 1e: Increase size for readability.

Fig. 2f: Add RA95 for comparison.

Fig. 2: Clarify the linear regression model based on AF2 prediction confidence.

Rework Fig. 3 legend to address issues regarding incorrect use of terms like preorganization, exceptional, and theozyme.

We highly appreciate this detailed feedback from the reviewer and adjusted all Figures for maximum clarity and readability. Since the overall structure and content of the initial manuscript has changed dramatically, Figure composition and captions have been redone with utmost care on the reviewer comments.

Referee #3:

In the manuscript at hand titled “Computational design of highly active de novo enzymes” Braun et al. present a strategy for computationally predicting enzymes using a newly developed machine-learning-based prediction algorithm called “Riff-Diff”. The authors claim that this new strategy can be in principle applied to create enzymes catalyzing various chemical reactions given the transition state of the reaction and an amino acid constellation that stabilizes that transition state. In the manuscript at hand, the authors focus on designing a (retro-)aldolase as a proof-of-principle. The designs resulting from their new design pipeline impressively generate catalytically active de novo enzymes within few numbers of designs (35 tested) and without the need for further directed evolution campaigns (“zero-shot”).

The Riff-diff design pipeline starts with a theozyme or catalytic array (a minimal set of amino acid residues that stabilize the transition state of the reaction) as an input. The Riff-Diff algorithm then aims to construct a protein backbone from helical peptide fragments which already place the catalytically relevant amino acid residues precisely in the catalytically active geometry (termed preorganization). A key step in the pipeline is the generation of an “artificial motif library” by combining rotamer-varied fragments into “motifs”, selecting them for physically sensible motifs in terms of compatibility of the individual rotamers with their backbone and avoiding steric clashes between fragments, and finally ranking the motifs according to the probabilities of the rotamers involved. The selected motifs are then further modified with an additional helical placeholder-fragment, which serves to promote the formation of a tighter binding pocket in the next step. Next, RFdiffusion with a modified auxiliary potential is used to construct the remaining backbone scaffold, while preserving the fragments, followed by Rosetta FastDesign-based optimization of residue-substrate interactions, and ranking with ESMFold. In an iterative manner the top scaffolds are re-subjected to additional cycles of backbone optimization. After multiple optimization cycles CoupledMoves further refines protein-substrate interaction, and ProteinMPNN finalizes the structure. The final structures are then ranked using AlphaFold2 prediction.

The entire design pipeline has been compiled into a single script, minimizing user input and thus potentially making it an easy/easier-to-use tool. Using their script, the authors proceeded to design de novo retro-aldolases targeting the compound methodol. Final structures showed a mostly high agreement with the targeted catalytic site geometry (observed differences between model and determined structures at or below 1 Å) and favourable predicted properties. When tested experimentally, some of the designed enzymes showed remarkably high catalytic performance. Two designs specifically, RAD35 and RAD29, approached $5 \cdot 10^6$ -fold reaction rate acceleration and the latter even rivalled a previously evolved retro-aldolase in catalytic efficiency. That these activities were achieved without further engineering is already an impressive and (to the best of the authors' and our knowledge) ground-breaking achievement.

To correlate protein structure, dynamics and activity, crystal structures of four designs were obtained, and MD simulations following the catalytic residues were performed. Here,

discrepancies emerged that put the importance of high preorganization of catalytic residues and precise positioning into question.

While in principle the presented design pipeline can be applied to any reaction with a known transition state that can be stabilized by amino acid residues, this versatility was not shown within the manuscript itself. Furthermore, it has yet to be demonstrated that the remarkably high success rate for the enzyme designs do transfer to other reactions and catalytic mechanisms. The enzymatic model reaction used here, has already undergone extensive experimental optimization in previous works and the idealized transition state geometry is therefore well established. The impact of lesser optimized catalytic systems on the success rate would be an interesting property of this new design tool and should be addressed in future work.

First, we want to sincerely express our gratitude to referee #3 for the excellent summary and reading and our manuscript in such detail! To directly address the reviewer's concern about the versatility of RiffDiff, the revised version of our manuscript now includes designs for the Morita-Baylis-Hillman (MBH) reaction. To stress also the capability of RiffDiff to work with lesser optimized catalytic systems, we used two distinct active sites, described in the literature, catalysing the MBH reaction. For one of which, we again obtained enzymes with activities rivalling those of laboratory evolved enzymes for the same reaction. We are convinced that this additional data proofs that RiffDiff can be used as a general purpose enzyme design tool.

Overall, the work seems to deliver a valuable high-precision enzyme design tool. Predicting de novo enzyme structures with high activity would immensely facilitate the development of biocatalytic processes by reducing the amount of screening and directed evolution campaigns to a minimum. The work therefore represent an important step in supporting the development of greener alternatives to conventional chemical transformation and is of moderately high impact or high impact if the aforementioned versatility is proven in a later work.

We thank the reviewer again for seeing the potential of the presented work and emphasize again that we've performed design work on a separate reaction to highlight RiffDiff's versatility.

Below are additional major and minor comments that should be addressed:

Major points:

1. The manuscript did not include or link to the source code for the Riff-Diff combined script. This should be added in the final version of the manuscript as several questions and minor comments could likely be resolved with it.

We apologize for not providing the link in the main text. It has been added now to the main text directly as well as to the data availability statement.

2. Fig. 1d: The caption of this figure requires a more detailed description of the data shown. Additionally, either a legend should be added or different types of points should be explained in the caption (e.g. do the white square (presumably) represent the median or average? Do the horizontal lines represent the total range or something different?).

We agree that the caption was lacking detail. During the addition of further design data and results, the revised manuscript was restructured in large portions, which also affected Figures and Figure panels. What was Figure 1d, is now more extensively displayed in Figure S1, with an updated text description.

3. It is unfortunate that no crystal structure could be obtained for the two most promising designs, RAD29 and RAD35. The conclusions drawn on p. 7, lines 29ff on precise residue placement thus rely primarily on the comparison of only one highly active design (RAD32) with three no-/low activity designs. Furthermore, as the authors point out themselves, the geometry RAD32 deviates from the intended design as the tyrosine of the tetrad has been unintentionally replaced by another tyrosine at a different position (a circumstance that to a lesser extent reminds of a criticism the authors made themselves in an earlier passage on p. 2, lines 40ff). Therefore, the conclusions on the importance of precise residue placement are a little weak and speculative and should be treated as such. Further experimental and computational work would be needed to elucidate the underlying causes as the authors correctly mention. This, however, might be out of the scope of the manuscript.

The paragraph now reflects the weakness of apo-state RMSD as an evaluation criterion of "precision". We thank the reviewer for pointing this out! To further substantiate our findings, however, we performed more in-depth analysis using further MD simulations (20 replicates) and additional AlphaFold3 predictions including the substrate. All of the findings are now summarized in a separate paragraph with the subheading 'Active site dynamics reveal uncoupling of precision and positioning'

4. The rationale to use a selection of rotamers for library building was to promote preorganization (p. 3, line 21). Yet, one of the final conclusions is that preorganization does not seem to be a critical factor for increased activity. Additionally, preorganization in the different designs is also only achieved to varying degrees. This unexpected finding raises the question which of the many considerations in the RiffDiff pipeline is critical for the high success rate or whether it is a complex interplay of all components. This warrants further discussion.

This is an excellent observation, and we thank the reviewer for raising this question. In the revised version, we detail the rationale for selecting backbone-compatible rotamers during library building and added a detailed description of it to the beginning of the supplementary materials. However, we agree that the high fraction of active designs produced by RiffDiff is a complex interplay of multiple components, which with the current datasets can only partially be pinned down. We provide further analysis and discussion at the end of the 'Active site dynamics reveal uncoupling of precision and positioning' paragraph and the conclusions.

5. The kinetic characterization of the designed retro-aldolases were performed in technical triplicate according to the caption of Fig. S6. The kinetic characterization of at least those designs that are critical for the findings in the main text (i.e. RAD29, RAD35, RAD13, RAD17, RAD36, RAD32) should also be validated as a biological duplicate (e.g. new, independent protein expression batch). Furthermore, the kinetic parameters reported in Table S3 should include measures of uncertainty (standard deviation or confidence intervals).

As suggested, we produced biological replicates of RAD variants and repeated the measurements. We emphasize that we included standard deviations or confidence intervals for all measurements concerning kinetics.

Minor points. A reaction scheme would be nice to include for the retro-aldol reaction of methodol.

Great point which we agree 100% with! We included a figure of the reaction schemes in the main text.

Page 4, lines 6-12: The placement of the auxiliary potential and placeholder helix is described rather cryptically.

Thank you for pointing this out! In fact this was pointed out by all referees and we now included a detailed description of the auxiliary potential and placeholder helix in the methods section and an overall very detailed description of RiffDiff at the beginning of the supplementary materials.

Page 4, lines 14-19: How many iterative cycles are used for scaffolding or is the number based on some criterium?

We apologize for not stating this explicitly. In the revised version, we now state that the number of refinement cycles is 5 (revised main text).

Page 4, line 30: The phrasing that AlphaFold2 ranks using “a set of metrics” is again very vague and might be addressed.

This is indeed very vague. We added more detail to the sentence directly and included a detailed description of metrics in the methods section.

Page 4, lines 44ff: Ambiguous phrasing: pLDDT values correspond to individual residues and the average pLDDT corresponds to the pLDDT values over a given structure. The value(s) given by the authors, is then presumably the average of the whole-sequence averages? Please clarify phrasing and also already cross-reference the corresponding swarm plot in Fig. 1d.

Thank you for pointing this out! To avoid any confusion, we changed pLDDT to average pLDDT, where applicable.

Page 5, lines 11ff: Why were 36 sequences chosen? How many designs had to be discarded due to the post-design criteria? How was the accessibility of the binding pocket assessed? The ambiguity in the text suggests it was through manual inspection. Please clarify.

This is an excellent question, and we thank the reviewer for reading and analysing our manuscript in such detail. In the revised version, we tried to clarify what was inspected manually and eventually modified in selecting the enzymes. Mostly, this inspection is concerned with identifying and reverting 'blocking' or 'occluding' mutations to the substrate entrance channel.

The exact number of 36 designed sequences originates from the layout of the expression screen, which uses 96-well plates. Thus with 36 sequences plus all the positive, negative and empty, buffer controls we could conveniently perform screening in duplicates.

The designs chosen for experimental validation were likely also influenced on the final ranking of the design pipeline. Does that ranking correlate with activity or any other experimental or structural property?

We asked ourselves the same question and mention several metrics related to structural quality correlated with chemical denaturation melting points. We could even fit a regression model to these data (Figure 3c and Supplementary Figure S9). In the revised manuscript, we also included a section about correlation of several AlphaFold3-derived metrics (which were not known at the time of design selection) with activity. We note though that the low sample size (35) limits the value of general correlational analyses on large sets of computational metrics at once.

Page 5, line 20: For the Michaelis-Menten experiments mentioned in this line, were these separate assessments or the same described on p. 6, lines 18ff? If so, please clarify phrasing (e.g. cross-referencing the paragraph). Mentioning the Michaelis Menten experiments here implies that batch-purification of the proteins has already happened, which is somewhat confusing as the next paragraph starts with the assessment of the that very batch purification. Please consider rephrasing for improving clarity.

Thank you for pointing this out. We rephrased the paragraph as suggested.

Fig. 2a: Conditions/instrumentalization of SEC analysis is missing. Please add these in the methods section or in the supplementary information. Furthermore, How were monomer elution times determined/ how was the instrument calibrated? At least one of the proteins (RAD30) seems to elute later than the others. Was it significantly shorter/ smaller than the other constructs? Providing the sequences as pointed out in the major comments, might be helpful to understand this.

We included a table containing sequences for all RADs in the supplementary materials. Conditions for SEC analysis are now included in the method section (in the gel filtration purification). We did not use a calibration curve with known molecular weight standards for the column used, but the combination of similar elution volumes and SAXS data confirms the

monomeric state of our designs. We do not have detailed explanation why RAD30 elutes at volumes corresponding to smaller size, even though mass spec analysis confirmed the correct molecular weight. Anecdotally though, for highly positively charged de novo proteins, we saw similar behavior in the past.

Fig. 2b: The label of the y-axis is a little confusing. If the y-axis depicts the “loss” of the 220 nm signal, the axis labels should start at 0, not 100% (in other words, there is no loss without heat treatment).

We updated the figure y-axis label. Thanks for alerting us to this!

Fig. 2c and p. 5, line 30: The text states that proteins were produced in high yields, implying that this is true for the majority of the designs. However, the majority of protein yields is below 20 mg per litre of culture, which can be considered moderate yield, with three designs having good yields of ~30 mg/L and above.

We agree that this is a misleading statement and changed the corresponding lines in the revised main text.

Fig. 2c: An additional legend or explanation in the caption is needed with regards to different elements in the figure (see also major comment for Fig. 1d). Additionally, is there a reason for RAD5 to be such an outlier?

The whole Figure was remade for the revised version. While we do not have a detailed explanation for the high χ^2 value of RAD5, a simple explanation is that either of the N- or C-terminal helix unfolds or refolds, thus providing a drastically different shape.

Please provide more detail on the linear regression model in Fig. 2d/ Fig. S5 in the captions or text.

In the revised version, this panel is now in Figure 3c. We added additional explanations to the caption and the Figure S9.

Fig. 2e caption: The caption mentions the “Schiff base intermediate”, while in reality it is the conjugate acid of the Schiff base (the iminium ion, this is also shown in 2e).

This is correct. Thank you for pointing this out. We changed the name of the intermediate state to the correct term, hemiaminal, where appropriate.

Fig. 3c: It is not entirely clear which distance is depicted here. Is the displayed distance in reference to the crystal structure as in Fig. S9 or was it calculated as in Fig. S10? Please add a brief explanation to the caption.

In the revised version of the manuscript, this panel entirely, as we repeated MD simulations with a higher number of replicates (20) and provide a much more in-depth analysis, with refined and careful statements about the active site rigidity.

Page 6, line 19: Michaelis-Menten parameters were determined for 30 (not 31 as written) of the retro aldolases according to Table S3 (5 were expressed but not determined, 1 did not express).

We corrected this in the main text. Thanks for pointing out this error!

Fig. S10: The colour-coding of the frames is not compatible with colour-blind inclusive design. I suggest replacement with or addition of an additional visual aid (e.g. symbol in the corner of the frames).

Figure S10 is replaced with a set of other Figures in the revised version of the study. Since we performed an additional 20 replicates of MD simulations, we could perform a much more in-depth analysis of active site flexibility. The total scope of the analysis is now split between several Figure panels in the main text and supplementary materials (Figure 4c and S16). We believe that this additional analysis strengthens the findings of our study.

Fig. S10: Please write out the abbreviated “NZ position” once in the caption.

We made sure that all abbreviations are spelled out at least once in the text. However, in this particular case that Figure has been removed entirely.

Methods: Please include the name of the plasmid which was used for cloning and in case of a not widely used plasmid, please include an entry number from a database/repository (e.g. Addgene). If a custom plasmid was used, it is recommended to submit the annotated sequence as an additional file or submitting it to a common plasmid repository and citing the entry. If it is not self-evident from the plasmid sequence, please also state which Golden Gate restriction sites were used.

We uploaded all data to Zenodo and provide a download link containing the sequence and plasmid map in the methods section.

Methods: When reporting centrifugation steps (e.g. p. 11, line 1), the acceleration (in rcf or g) should be reported, not the rpm number because the latter is instrument-dependent. Duration and temperature of the centrifugation step should also be included. In the subsequent sentence about washing the pellet some information is missing, e.g. at which temperature were the pellets washed, how many times with which volume?

Thank you for pointing this out! We changed rpm to rcf, where appropriate. In addition, we updated the information about washing the pellets.

Methods, p. 11, lines 23ff: Were the same protein concentrations used for the chemical denaturation experiments as in the previous paragraph about the thermal denaturation?

We apologize for not explicitly stating the concentrations and updated the missing information about the chemical denaturations in the Methods section

Methods: The Python SciPy library was used (e.g. p. 11, line 28/p. 12, line 32). Please include a version number.

The exact version numbers were added to the Methods section.

Page 13, line 23: The number after 6-methoxy-2-naphthaldehyde “1” is probably for numbering the compound (but no other compound is numbered). If a reaction scheme is added (see minor comment above), additional compound numbers would be useful and should be formatted differently (e.g. in bold font).

Excellent suggestion! We added a reaction scheme and numbering of chemical compounds.

Page 13, line 24: Was the phosphate buffer's pH adjusted to a specific value?

Yes - we specified the adjusted pH (7.4) for all phosphate buffers.

Page 7, line 1ff: Here, the enantiomeric preference of only one design was investigated. What are the enantiomeric preferences for other constructs, e.g. the other highly active RAD35 or the active designs from which crystal structures could be obtained? Furthermore, the retro-aldol reaction was performed only with a racemic substrate – were different cleavage rates observed for the individual enantiomers (i.e. kinetic resolution of the racemic mixture)?

This is a great question, and we specifically performed a whole set of additional experiments to clarify it. The revised manuscript now contains a new main text paragraph with the subheading ‘RAD29 and RAD35 are highly stable and stereoselective biocatalysts’. We believe that this adds valuable context to the whole study and would like to thank the reviewer again for asking the question.

Page 9, line 23ff: Please include a reference for CASP.

The section talking about contest like CASP has been removed in the revised version.

Supplementary information: Please provide the 36 sequences chosen for the study in form of a table/ additional excel file or other adequate file format. Please include also sequences and used annealing temperatures of primers from PCR experiments. The annealing time for the PCR reaction is missing in the methods section.

The missing information was added. Tables for sequences and primers (including annealing temperature) can now be found in the supplementary materials.

Referee #1:

The authors have clearly invested substantial effort in addressing the reviewer comments. They have added significant additional information and analyses (e.g., MD equilibration) which have considerably enhanced both the quality and comprehensiveness of the work. Additionally, the authors have improved the structure and documentation of the associated GitHub repository, in this way supporting reproducibility and usability of their in-silico workflows. I recommend the article for acceptance, subject to the following concern:

I strongly encourage the authors to remove the “original” and insufficiently documented RiffDiff repository (https://github.com/mabr3112/riff_diff_original), as it may cause confusion for users. Alternatively, they should clearly link to the better-documented repository (https://github.com/mabr3112/riff_diff_protflow).
Answer: We agree and thus changed the visibility of the riff_diff_original repository to *private* to avoid confusion for potential users and removed references to it from the manuscript.

Answer: We agree and thus changed the visibility of the riff_diff_original repository to *private* to avoid confusion for potential users. We also removed references to it from the manuscript.

Moreover, it is potentially problematic that the code is only made available under the umbrella of a new, previously undescribed framework (“ProtFlow”), which is neither introduced nor discussed in the current manuscript. The RiffDiff code should be presented as a standalone, clearly defined methodological workflow.

Answer: We added detailed instructions on how to run RiffDiff to the riff_diff_protflow github repository. In addition, our python library ProtFlow comes with its own repository and extensive documentation. We believe that this type of documentation is sufficient for anyone to effectively use both.

In my view, the current mixture of acronyms, GitHub repositories, and workflow frameworks could hinder broader adoption of RiffDiff. Clarifying this aspect would substantially improve the overall accessibility and impact of the work.

Answer: Please see explanation above

Referee #1 (Remarks on code availability):

Please refer to general review comments.

Referee #2:

The authors have satisfactorily addressed most of my concerns; however, a few minor issues remain that should be addressed to improve readability and accuracy. The addition of MBHase designs has strengthened the paper, but some minor revisions in the description and interpretation of those results are needed for greater clarity. I recommend publication after all of the following minor comments are addressed:

Answer: We thank the reviewer for their favorable evaluation, specifically for their interest in the MBH designs.

Comment 1.

Abstract:

The authors claim that their “findings enable the practical applicability of de novo protein catalysts in synthesis and shed light on fundamental principles of protein design and enzyme catalysis.” However, the practical applicability of their enzymes for synthesis is not convincingly demonstrated. The retro-aldolases are not synthetically useful, and while the MBH reaction is valuable, the reported conversions ($\leq 16\%$) are too low to be considered practically useful. I recommend omitting this claim

and rewording the abstract to instead highlight the key achievements: the design of de novo enzymes for two distinct reactions with higher activity than previous de novo enzymes, achieved by constructing a custom scaffold for the catalytic array rather than repurposing a natural scaffold.

Answer: We rephrased the sentence from “These findings enable the practical applicability of de novo protein catalysts in synthesis” to “pave the way towards the practical applicability of de novo protein catalysts in synthesis” to weaken our claim.

Comment 2.

Page 2, line 32:

The statement “Since the first successful attempts, critical analysis of designed enzymes and their evolved variants revealed additional aspects of the enzyme design problem. One was the construction of catalytically potent theozymes” is unclear, as the term “catalytically potent theozyme” is not defined. A clearer phrasing might be:

“Since the first successful attempts, critical analysis of designed enzymes and their evolved variants revealed key challenges in enzyme design. One such challenge was the difficulty in predicting optimal theozymes for the reaction of interest.”

Answer: We thank the reviewer for this suggestion as it defines the problem much clearer. We modified the corresponding sentence accordingly.

Moreover, the term “catalytically potent theozyme” remains unclear. What does it mean for a theozyme to be “potent”? Rather than referring to potency, it would be more accurate to state that the original theozymes used for the retro-aldol and MBH reactions were suboptimal, as evidenced by the fact that directed evolution replaced the designed catalytic arrays with alternative residues that enhanced catalytic efficiency.

For reference, I include below my original comment on this point:

“The term “catalytically potent theozyme” is also unclear. Is this potency based on theozyme DFT energy? Likely, the authors mean that the catalytic tetrad motif was empirically determined to be more efficient for the retro-aldol reaction than simpler motifs, such as the Lys/Glu pair used by Althoff et al. (Prot. Sci., 2012) to design RA95. Additionally, the tetrad with bound inhibitor used here is not a theozyme, as it is not a QM model of catalytic groups stabilizing a transition state. I recommend that the authors verify whether they are properly using the term theozyme throughout their article (see Tantillo et al., Curr. Opin. Chem. Biol., 1998).»

Answer: We thank the reviewer for this comment and agree that our phrasing is mixing up different things. Thus, we modified the sentence about “catalytically potent theozymes” according to the suggestion.

Comment 3.

Page 2, Line 23:

Please cite Privett et al. (<https://pubmed.ncbi.nlm.nih.gov/22357762/>), in addition to Siegel, Rothlisberger, and Jiang. It’s important to acknowledge other pioneers in the field beyond just David Baker, including Steve Mayo.

Answer: We agree and added the citation.

Comment 4.

Page 3, Line 2:

The authors again describe “preorganization” incorrectly when they mean flexibility: “molecular dynamics simulations of retro-aldolases at intermediate steps during directed evolution demonstrated the importance of preorganization in catalytic arrays.” Preorganization and flexibility are not the same thing. This sentence should be reworded to avoid confusion.

For context, I paste below my original comment related to this topic:

“Preorganization Misinterpretation. The authors use MD to evaluate catalytic residue preorganization but MD, as applied here, cannot measure this. Preorganization refers to active-site groups not having to undergo significant rotation to stabilize the transition state, since they are already pointing in the correct direction (see Jindal & Warshel, *Proteins*, 2017). This can only be shown by comparing active-site structures with and without a transition state (or analogue). Here, unbound structures were analyzed, so preorganization cannot be determined, as conformational changes may occur upon transition-state formation. The authors instead have assessed rigidity of the catalytic tetrad, indicating its ability to adopt productive or unproductive conformations, given the assumption that the designed configuration is catalytically productive. The authors should thus remove all discussion of preorganization from the manuscript and focus on rigidity/flexibility and/or productive/unproductive conformations. Along the same lines, on Fig. S9, it is stated that the catalytic tetrad is well preorganized. Notwithstanding the inability of MD as performed here to assess preorganization, the Y180 residue is shown to be highly flexible, which presumably would not be ideal for efficient catalysis.»

Answer: We apologize for not clarifying this further in our first revision. The particular section that is referenced above is talking about the evolution of RA95 and there have indeed been publications showing enhanced preorganization using MD simulations with reaction intermediate states for this. We added the corresponding citations. However, we removed the explicit mention of preorganization to shorten the respective chapter.

Comment 5.

Page 7, Line 4:

To facilitate analysis by reader, please indicate the fold-decrease in *k_{cat}* caused by mutation of tetrad residues (e.g., “Substitutions reduced *k_{cat}* by X–Y fold”).

Answer: We changed the text, as suggested.

Comment 6.

Page 7, Line 34 and Fig. 3c:

Specify the linear regression model used and provide the equation. This is necessary for reproducibility.

Answer: We apologize for the missing explanation of the regression model. This is now included in the Supplementary Materials.

Comment 7.

Page 8, Line 25:

Replace “can” with “could.”

Answer: Done.

Comment 8.

Page 8, Line 31:

The unproductive conformation of Y120 is unlikely to result from crystal packing, as this residue is located within the active site rather than in a surface-exposed region where crystal contacts typically occur. Therefore, it is unlikely to be a crystallization artifact, as suggested in the text. More likely, this reflects an inaccuracy in the design model.

Answer: This is speculative, and we thank the reviewer for pointing this out. We agree that the unproductive conformation of Y120 observed in the crystal structure is unlikely to stem from crystal packing artifacts, as the residue is far from the surface. However, since the Y120A variant decreased activity, we believe that Y120 must adopt the designed orientation during catalysis but might have a stronger preference for the observed rotamer from the crystal structure. This could also be related to the cryo-temperature at which the diffraction dataset was recorded or in the absence of substrate. We rephrased the entire segment and now focus on the importance of

catalytic contributions beyond the interactions specified in the catalytic array instead of vague explanation attempts.

Comment 9.

Page 9, Line 22:

Please explain what is meant by "active-site metric". This term is unclear and should be defined.

Answer: Thank you for pointing this out! We added an extensive description of the active site metric and all equations to the Supplementary Materials.

Comment 10.

Page 9, Line 36:

The phrase "Streptavidin cofactor" is incorrect. Streptavidin is a protein, not a cofactor. Please clarify or revise this terminology.

Answer: We apologize for this error and are sorry that we overlooked this. We changed the incorrect "Streptavidin cofactor" to "artificial cofactor embedded in streptavidin".

Comment 11.

Page 10, Line 1 and Abstract:

The terms "active site constellations" and "active site arrays" are used interchangeably. Please standardize to one term throughout. I recommend "array", as "constellation" is not as common in enzymology literature.

Answer: We highly appreciate the level of detailed review and agree that one term should be used throughout. We changed each mention of "active site constellation" to "active site array".

Comment 12.

Page 10, First Paragraph:

Please add citations to Crawshaw (Ref 51) and Hutton (Ref 52) at the appropriate points in the discussion where their work is referenced.

Answer: We added the citations where the work is referenced.

Comment 13.

Suppl. Fi. S10:

Label each transition state (TS) and intermediate. This will help match the text (first paragraph on Page 10). Also, please explicitly show the MBHase mechanism involving Glu26 as acid/base in a separate panel to help readers clearly distinguish the two MBHase mechanisms discussed here.

Answer: Great suggestion! We updated the figure to include the reaction mechanism for BH1.8 and included labels for each intermediate and transition state.

Comment 14.

Page 10, Line 17:

Please indicate which transition state was used in the RiffDiff design of MBHases (refer to Suppl. Fig. S10).

Answer: We included a reference to Suppl. Figure S11b when describing which transition state model was selected. In the original publication, this transition state is called TS3 (from Int2H to Int3). Since we want to use consistent numbering of intermediates (INT1 to INT4) and the stepwise deprotonation is more likely than a concerted mechanism according to Crawshaw et al., we updated the transition state numbers. Accordingly, the selected transition state is now called TS4.

Comment 15.

Page 10, Line 23:

Please provide geometric definitions (distances, angles, dihedrals) for the interactions between catalytic residues and TS (e.g., H-bonds, π - π stacking interactions, etc.) as a Supplementary Table or Figure.

Also, please provide the cartesian coordinates for the theozymes used (e.g., in mol2 or PDB format) as Supplementary Files. This is required to enable others to reproduce the work.

Answer: We thank the reviewer for this suggestion. Rather than extracting and tabulating all geometric parameters for the interactions between catalytic residues, we have provided the full atomic coordinates of all catalytic arrays in PDB format as a Supplementary Data set, available via Zenodo (<https://doi.org/10.5281/zenodo.15979364>). These files allow detailed analysis of distances, angles, and dihedrals using commonly available molecular visualization and analysis tools such as PyMOL. We have added a note to the manuscript to indicate the availability of these files for reproducibility.

Comment 16.

Page 11, First Paragraph:

MBH48 should be compared directly to BH1.8 23H (which shares the same catalytic array), in addition to BH32.8 (which uses a different array). For example:

“With a k_{cat} of 0.025 min^{-1} , MBH48 is 1.5 times more active than BH32.8 ($k_{cat} = 0.0168 \text{ min}^{-1}$), a variant that emerged after screening 13,590 clones over 8 rounds of directed evolution (Figure 5e). However, it remains 45-fold less active than BH1.8 23H ($k_{cat} = 1.13 \text{ min}^{-1}$), which shares the same catalytic array.” This comparison is critical given the paper’s emphasis on “catalytically potent theozymes” and underscores the need for future designs to incorporate features beyond precise catalytic array positioning. It also provides a more balanced perspective on the limitations of the current approach.

Answer: We included the comparison to BH1.8 23H activity in the text and thank the reviewer for this suggestion, which we agree provides a much better perspective on the approach.

Comment 17.

Discussion and Outlook

This section contains several statements without appropriate citations. The discussion should situate the findings more clearly within the existing literature. Please add citations for the key statements listed below:

Page 11, Line 35: “Current enzyme design and engineering rely on high-throughput screening methods to produce viable enzymes.”

Answer: We edited the entire discussion section to improve conciseness and have made sure that appropriate citations are present where work of others is referenced.

Page 12, Line 23: “Our findings support the notion that designing enzymes with activities similar to their natural counterparts will require accounting for catalytic interactions and their conformational dynamics throughout the complete reaction cycle.”

Answer: Please see above.

Page 12, Line 27: “Protein language models represent a new frontier for this task, but their poor generalizing abilities currently limit their effectiveness in novel chemical reactions, especially with de novo enzymes.”

Answer: Please see above.

Page 12, Line 29: “Physics-based methods like QM/MM appear more suited to handle generalizability.”

Answer: Please see above.

Page 12, Line 31: “An alternative route for physics-based activity prediction is conformational ensembles of the active site.”

Answer: Please see above.

Page 12, Line 36: “This highlights how computational tools that optimize an active site’s conformational ensemble could yield future improvements in activity of designed enzymes.”

Answer: Please see above.

Page 13, Line 7: “Yet very few of these studies started from scratch and all those that did relied on directed evolution or screened several hundred de novo sequences to reach practically useful activities in the designed biocatalysts.”

Answer: Thanks for pointing this out. We rewrote the entire paragraph. This section now reads: With the ability to robustly scaffold catalytic arrays, Riff-Diff offers a way to go beyond screening campaigns and help transform enzyme design from an orphan field into an approach that the broader biotechnological community can apply.

Comment 18.

Please make Suppl. Fig. S5 bigger, including fonts. It is hard to read.

Answer: Good suggestion! We split the figure into two supplementary figures and increased the size of both.

Comment 19.

Fig. 3f: ee values in bar graph don't match the legend.

Answer: Thank you for alerting us to this confusing point! We rearranged the figures slightly and exchanged figures 3f and 3e. The ee% values mentioned in the legend of 3f are of the (unreacted) substrate methodol when running the reaction in the retro-aldol direction. These ee% values are not displayed and do not correspond to the values displayed in 3f, which are the enantiomeric excess of product methodol when running the reaction in the aldol direction. We updated the figure legend and believe that it is much clearer now.

Comment 20.

Fig. 4: Structures are very similar but not almost identical (1.2 Angstrom RMSD is not almost identical). Please reword.

Also, the figure legend describes a theozyme for panel c but no theozyme is shown, only catalytic array.

Answer: We appreciate the level of detail of the review very much and exchanged “almost identical” with “closely resemble” to tone down the statement and “theozyme” with “catalytic array” to make it more correct. However, there is a deeper argument here that needs further discussion that is unfortunately beyond this manuscript. 1.2 Angstrom RMSD over all C-alpha atoms is in our opinion virtually identical. It is below the average RMSD in a standard NMR ensemble of a protein structure and within the margin of error of building a model into an electron density map of this resolution.

With these revisions, the manuscript will present its findings more clearly and accurately and will better position itself within the context of existing work in enzyme design and catalysis.

Answer: We agree and want to thank Referee #2 again for the great feedback on several points!

Referee #2 (Remarks on code availability):

It works. We are using some aspects of it in our research.

Answer: Fantastic! That was the goal.

Referee #3:

The authors have addressed nearly all of my previous comments to full satisfaction and even exceeded expectations in answering remaining questions. Especially with the additional challenging application on the abiological Morita-Baylis-Hillman reaction, the general applicability and thus potential impact of the new de novo enzyme prediction pipeline is now well supported. I therefore fully support publication of the work.

Answer: We thank the reviewer for their kind words and are happy that we satisfactorily answered all their questions.

I recommended the following minor adjustment for the final manuscript:

1. One of my comments on the initial draft concerned the type of replicates (comment point 5), mainly to include biological duplicates for major constructs. The authors state in the reworked manuscript that biological duplicates were performed in the methods section when describing protein expression. In the rebuttal document the authors also say that the data from biological replicates is part of the analysis already in the reworked manuscript. The description of data (e.g. Michaelis-Menten analyses) do not clearly reflect this. In figure captions it is still only stated that the data stem from "triplicates", implying the figures represent single biological replicates of technical triplicates. The data of biological replicates can be included as supplemental material but should then be cross-referenced where needed. If the biological duplicates of the triplicate measurements are already part of the data, the figure descriptions should reflect this to avoid confusion.

Answer: We apologize for the confusion. The statement has been adjusted to reflect the technical triplicate measurement in the shown kinetics. We also included a reference to the supplementary table which shows the values for the biological duplicate measurements to the 'Activity Measurement' paragraph of the methods section.

2. Several figures (1c, 3d, 4c,d, 5e, S3 and S14a) are still very hard to interpret for people with colour vision impairment, which becomes apparent when viewed as a grayscale document. For the respective figures I suggest adapting to a colour-blind palette or in case of line charts, adding different texture to the lines (e.g. dotted lines, etc.).

Answer: We adjusted the figures according to the Referee's suggestion and selected color-blind friendly palettes and textured lines for our plots. Thank you for this important note!

3. Methods section in main document, p. 22, l. 4 typo: "p 7.4" instead of "pH 7.4"

Answer: We corrected the typo from "p" to "pH".

Referee #4:

I co-reviewed this manuscript with one of the reviewers who provided the listed reports.

Answer: We appreciate the detail of the review, which helped us to improve the manuscript a lot.

Point-to-point responses:

Referee #1:

I appreciate that the authors removed the old and poorly documented repository and agree that the “protflow implementation” of Riff-Diff is now well-documented enabling everybody from the target audience to execute the pipeline effectively.

Answer: We thank the reviewer for their favorable assessment and are happy that our efforts to provide detailed documentation is valued.

Referee #2:

The authors have addressed all my comments satisfactorily.

However, I noticed that the mass spectrometry methods are not included in the Methods section. Was the analysis performed by MALDI? Which instrument was used? Please provide the missing methodological details.

Answer: We apologize for not mentioning this explicitly earlier. We now added a separate paragraph in the methods section to explain the mass spectrometry analysis in detail.

Congratulations on your nice work!

Answer: We thank the reviewer for their favorable evaluation